# A novel targeting domain directs essential components of the cytosolic iron–sulfur cluster assembly pathway to the mitochondrion of *Toxoplasma* parasites

Evie R. Hodgson[¤], Jenni A. Hayward, Rachel A. Leonard, Fadzai Victor Makota, Giel G. van Dooren[ID]*

Research School of Biology, Australian National University, Canberra, Australian Capital Territory, Australia

¤ Current address: Department of Biochemistry and Pharmacology, University of Melbourne, Parkville, Victoria, Australia
* giel.vandooren@anu.edu.au

## Abstract

The assembly of iron–sulfur (FeS) clusters for cytosolic and nuclear proteins is essential for eukaryotic cell biology. This assembly is mediated by the Cytosolic Iron–sulfur Assembly (CIA) pathway, which localizes to the cytosol of most eukaryotes. We showed previously that the scaffold protein on which cytosolic FeS clusters assemble localizes to the mitochondrion of the apicomplexan parasite *Toxoplasma gondii*. The localization and importance of the remainder of the pathway in these parasites, however, remained unclear. Here, we undertake a comprehensive analysis of the CIA pathway in *T. gondii* parasites. We present evidence that the CIA pathway localizes predominantly to the mitochondrion of the parasite and is essential for parasite survival. We show that the three proteins that make up the CIA Targeting Complex (CTC), which facilitates the transfer of FeS clusters to cytosolic and nuclear client proteins, exhibit dual localization to the mitochondrion and cytosol. We reveal that mitochondrial targeting of the CTC is mediated by a novel loop on the CIA1 protein of the complex, and that this loop is critical for parasite survival. We show that an aromatic amino acid motif in the loop facilitates mitochondrial targeting, and that this loop is functionally conserved in apicomplexans and their closest free-living relatives. Our study provides a comprehensive analysis of the CIA pathway in an important group of intracellular parasites, and elucidates pivotal differences in an otherwise ancient and highly conserved biosynthetic pathway that may reflect an evolutionary fitness advantage conferred on *Toxoplasma* and related organisms.

which permits unrestricted use, distribution, and reproduction in any medium, provided the original author and source are credited.

**Data availability statement:** All relevant data are within the paper and its Supporting information files.

**Funding:** This research project was supported by grants from the Australian National Health and Medical Research Council (GNT1182369; https://www.nhmrc.gov.au/) and Australian Research Council (DP230100853; https://www.arc.gov.au/) to GGvD, an Australian Government Research Training Program scholarship to JAH (https://www.education.gov.au/research-block-grants/research-training-program), and a Gwendolyn Woodroofe PhD scholarship from the ANU to RAL (https://study.anu.edu.au/scholarships/find-scholarship/anu-gwendolyn-woodroofe-phd-scholarship-biological-sciences). The funders did not play any role in the study design, data collection, data analysis, decision to publish, or preparation of the manuscript.

**Competing interests:** The authors have declared that no competing interests exist.

**Abbreviations:** ATc, anhydrotetracycline; BN, blue native; BSA, bovine serum albumin; CIA, Cytosolic Iron-sulfur Assembly; CTC, CIA Targeting Complex; DMEM, Dulbecco's Modified Eagle Medium; FeS, iron-sulfur; GFP, green fluorescent protein; HA, hemagglutinin; HFFs, human foreskin fibroblasts; IAA, 3-indoleacetic acid; ISC, Iron Sulfur Cluster; PBS, phosphate-buffered saline; PVDF, polyvinylidene difluoride; SDS-PAGE, Sodium Dodecyl Sulfate-Polyacrylamide Gel Electrophoresis; TMD, transmembrane domain.

## Introduction

Iron–sulfur (FeS) clusters are among nature's most ancient yet versatile cofactors [1]. They serve as prosthetic groups for proteins involved in a myriad of fundamental cellular processes, such as in electron transport chains, ribosome assembly, DNA repair and synthesis, and iron homeostasis [1,2]. FeS clusters cannot be scavenged from the environment and, instead, require dedicated machinery to facilitate their biosynthesis. In eukaryotes, distinct biosynthetic pathways are expressed in the specific subcellular compartments that harbor FeS cluster-requiring proteins [3]. Mitochondrial FeS proteins obtain clusters synthesized by the so-called Iron Sulfur Cluster (ISC) pathway. The Sulfur Mobilization pathway, a similar but functionally distinct pathway, supplies clusters to plastid-localized FeS proteins. Finally, the so-called Cytosolic Iron–sulfur Assembly (CIA) pathway generates [4Fe-4S] clusters in the cytosol and distributes them to both cytosolic and nuclear FeS proteins.

The synthesis and maturation of cytosolic FeS clusters depend on the provision of a sulfur- (and possibly iron-) containing product from the mitochondrial ISC pathway [4,5]. Electrons required for the assembly of cytosolic FeS clusters are derived from NADPH via an electron transport chain consisting of the diflavin reductase, Tah18 (also referred to as NDOR1 or ATR3), and the FeS protein, Dre2 (also referred to as CIAPIN1; [5]). In most eukaryotes, the integration of iron, sulfur, and electrons into cytosolic [4Fe-4S] clusters occur on a scaffold complex consisting of the P-loop NTPases NBP35 and Cfd1 [6,7]. Once assembled, [4Fe-4S] clusters are thought to be trafficked by the Nar1 protein (also referred to as IOP1, NARFL or CIAO3) to the so-called CIA Targeting Complex (CTC; [8–11]). The CTC is comprised of the proteins CIA1 (also referred to as CIAO1), CIA2 (also referred to as CIAO2B, MIP18 or AE7), and MMS19 (also referred to as Met18) and facilitates the incorporation of FeS clusters into cytosolic and nuclear apo-FeS proteins, thereby serving as a bridge between the FeS cluster carrier Nar1 and recipient FeS proteins [6,10,12].

Apicomplexans constitute a large phylum of protozoan parasites that impart considerable health and socio-economic burdens globally. Apicomplexans are a part of the larger Alveolate superphylum of eukaryotes, which also include ciliates (e.g., *Paramecium tetraurelia* and *Tetrahymena thermophilus*), dinozoans (e.g., the photosynthetic coral symbiont *Symbiodinium microadriaticum* and the oyster parasite *Perkinsus marinus*), and a recently identified group known as the chrompodellids (e.g., *Chromera velia* and *Vitrella brassicaformis*). Within the alveolates, the apicomplexans, chrompodellids and dinozoans form a clade known as the myzozoans [13,14]. Key apicomplexan parasites include those of the *Plasmodium* genus, the causative agents of malaria in humans, and *Toxoplasma gondii*, the causative agent of toxoplasmosis. The core proteins involved in the CIA pathway, including all three proteins of the CTC, are conserved in apicomplexans [15]. We demonstrated recently that the *T. gondii* homolog of the CIA scaffold protein NBP35 (*Tg*NBP35) was essential for parasite survival [15]. In contrast to its cytosolic localization in most other eukaryotes, we found that *Tg*NBP35 was anchored to the outer face of the outer mitochondrial membrane courtesy of an N-terminal transmembrane domain (TMD). The N-terminal

TMD of *Tg*NBP35 is conserved throughout myzozoan NBP35 homologs, but absent from ciliates, where the NBP35 homolog localizes to the cytosol [15,16]. These findings suggest that the assembly of cytosolic [4Fe-4S] clusters occurs on the cytosolic face of the mitochondrion of myzozoans.

The subcellular localization and importance of the remaining CIA pathway proteins has not been studied previously in myzozoans, and we set out to test this using *T. gondii* as our experimental model. We found that most candidate proteins of the *T. gondii* CIA pathway localize to the parasite mitochondrion and are critical for parasite proliferation. Curiously, we found that the CTC proteins *Tg*CIA1, *Tg*CIA2, and *Tg*MMS19 exhibit a dual localization to the mitochondrion and cytosol, with mitochondrial localization of this complex mediated by a loop on the *Tg*CIA1 protein that is conserved throughout myzozoans, and which is critical for its role in parasite biology. Taken together, our study reveals that the mitochondrion acts as a hub for the CIA pathway in *Toxoplasma* and related organisms.

## Results

### Cytosolic FeS cluster assembly occurs at the mitochondrion of *Toxoplasma gondii*

We set out to determine the subcellular localization of the previously uncharacterized candidate CIA pathway proteins in *T. gondii*. Using CRISPR/Cas9 genome editing, we introduced a 3× hemagglutinin (HA) epitope tag into the 3′ end of the coding sequences of the proposed electron transfer chain proteins *Tg*Tah18 (www.ToxoDB.org gene identifier TGGT1_249320; [17]) and *Tg*Dre2 (TGGT1_216900), and all three of the proposed CTC proteins, *Tg*CIA1 (TGGT1_313280), *Tg*CIA2 (TGGT1_306590) and *Tg*MMS19 (TGGT1_222230), in an anhydrotetracycline (ATc)-regulatable *Tg*NBP35-cMyc strain that we had generated previously [15]. We validated correct integration of the tag in each parasite line by PCR analysis (S1A–S1E Fig). We were unsuccessful in integrating a HA tag into the 3′ end of the coding sequence for the proposed FeS transfer protein *Tg*Nar1 (TGGT1_242580). Instead, we introduced a HA-mini-Auxin Inducible Degron (HA-mAID) tag at the 5′ end of the coding region of *Tg*Nar1 in a tdTomato and Tir1-expressing parasite strain (RH∆*ku80*/Tir1-FLAG/tdTomato; [18]; S1F Fig). To test whether each protein was expressed in the disease-causing tachyzoite stage of the parasite life cycle, we extracted proteins from the *Tg*Tah18-HA, *Tg*Dre2-HA, HA-mAID-*Tg*Nar1, *Tg*CIA1-HA, *Tg*CIA2-HA, and *Tg*MMS19-HA lines, separated them by Sodium Dodecyl Sulfate-Polyacrylamide Gel Electrophoresis (SDS-PAGE), and performed western blotting with anti-HA antibodies. All six proteins were expressed and were of approximately the expected masses for the epitope-tagged proteins (Fig 1A–1F).

To determine the subcellular localization of each protein, we performed immunofluorescence assays. We found that *Tg*Tah18-HA and HA-mAID-*Tg*Nar1 both co-localized with the mitochondrial marker *Tg*Tom40, whereas *Tg*Dre2-HA localized throughout the parasite cytosol (Fig 1G–1I). Curiously, all three proteins of the putative CTC exhibited a dual localization, overlapping with *Tg*Tom40 in the mitochondrion and also exhibiting diffuse labeling through the cytosol (Fig 1J–1L). Taken together, these data indicate that all the putative CIA pathway proteins in *T. gondii*, with the exception of *Tg*Dre2 but including the previously characterized scaffold protein *Tg*NBP35 [15,16], localize to the mitochondrion. We showed previously that *Tg*NBP35 localizes to the cytosolic face of the outer mitochondrial membrane [15], and our data are therefore consistent with the CIA pathway occurring on the cytosolic face of the outer mitochondrial membrane in *T. gondii* (Fig 1M).

### The CIA pathway is critical for parasite proliferation and protein synthesis in *Toxoplasma gondii*

We set out to test the importance of each of the candidate CIA pathway proteins for parasite proliferation. Using CRISPR/Cas9 genome editing, we introduced mAID-HA epitope tags at the 3′ ends of the coding regions for *Tg*Tah18, *Tg*Dre2, *Tg*CIA1, *Tg*CIA2 and *Tg*MMS19 of tdTomato/Tir1 parasites (S2 Fig). The mAID tagging system facilitates proteasomal degradation of mAID-fused proteins upon the addition of the small molecule 3-indoleacetic acid (IAA; [19]). Since the abundance of the resulting proteins can be conditionally regulated, we termed the resulting lines expressing mAID-tagged CIA pathway proteins regulatable (r)-*Tg*Tah18-mAID-HA, r*Tg*Dre2-mAID-HA, rHA-mAID-*Tg*Nar1, r*Tg*CIA1-mAID-HA, r*Tg*CIA2-mAID-HA, and r*Tg*MMS19-mAID-HA. To determine whether the mAID-tagged proteins could be regulated upon

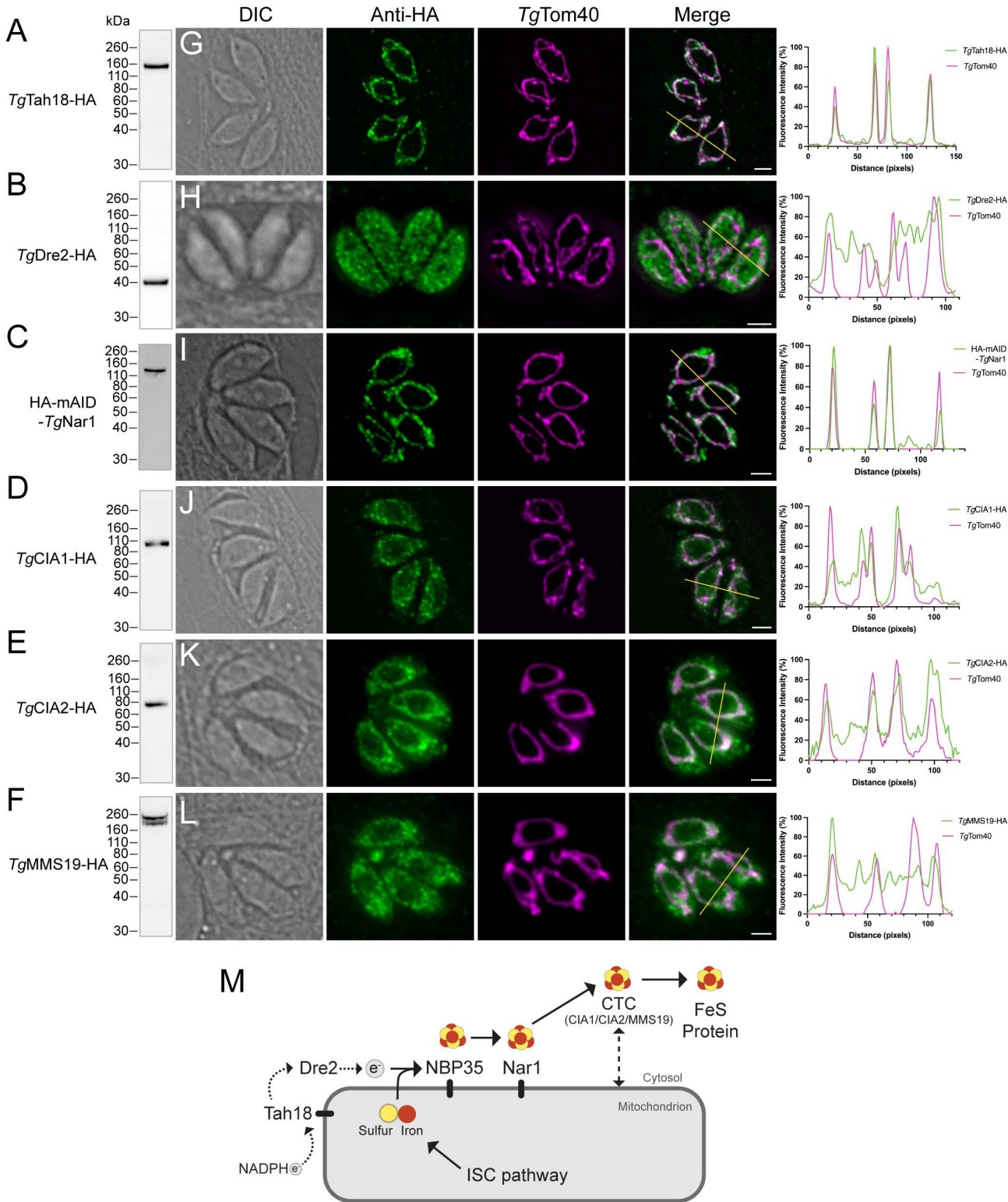

**Fig 1. The CIA pathway localizes to the mitochondrion of *Toxoplasma gondii*. (A–F)** Western blots of proteins extracted from **(A)** *Tg*Tah18-HA, **(B)** *Tg*Dre2-HA, **(C)** HA-mAID-*Tg*Nar1, **(D)** *Tg*CIA1-HA, **(E)** *Tg*CIA2-HA, or **(F)** *Tg*MMS19-HA expressing parasites, separated by SDS-PAGE and probed with anti-HA antibodies. *Tg*MMS19-HA appears as two separate bands, the lower of which may represent a degradation product. **(G–L)**

Immunofluorescence assays of **(G)** *Tg*Tah18-HA, **(H)** *Tg*Dre2-HA, **(I)** HA-mAID-*Tg*Nar1, **(J)** *Tg*CIA1-HA, **(K)** *Tg*CIA2-HA, or **(L)** *Tg*MMS19-HA expressing parasites, probed with anti-HA antibodies (green) to label the proteins-of-interest and anti-*Tg*Tom40 antibodies (magenta) to label the mitochondrion. Scale bars are 2 μm. DIC, differential interference contrast transmission image. Right, corresponding fluorescence profiles depicting the intensity of anti-HA (green) and anti-*Tg*Tom40 (magenta) labeling along the yellow line on merged images. The numerical data underlying this Figure can be found in S1 Data. **(M)** Model for the CIA pathway in *T. gondii* parasites, with Tah18, NBP35 and Nar1 positioned on the outer face of the mitochondrion, Dre2 in the cytosol, and the CIA Targeting Complex (CTC; consisting of CIA1, CIA2 and MMS19), exhibiting dual cytosolic and mitochondrial localization. Electrons (e⁻) are sourced from NADPH via an electron transfer chain consisting of Tah18 and Dre2. Sulfur (yellow) and possibly iron (red) are sourced from the iron–sulfur cluster (ISC) synthesis pathway in the mitochondrion, assembled as [4Fe-4S] clusters on the NBP35 scaffold, then trafficked via Nar1 to the CTC. The CTC donates FeS clusters to client FeS proteins.

the addition of IAA, we cultured each parasite line in the absence or presence of IAA for 24 h. We extracted the resulting proteins, separated them by SDS-PAGE and performed western blotting. We found that the abundances of all six candidate CIA pathway proteins were substantially depleted upon IAA addition (Fig 2A–2F).

To explore the contribution of each CIA pathway protein to parasite proliferation, we performed fluorescence-based proliferation assays, measuring tdTomato fluorescence in parasites across time as a function of parasite proliferation. We found that in the absence of IAA, all parasite lines proliferated normally, exhibiting a typical sigmoidal growth curve (Fig 2). The parental RHΔ*ku80*/Tir1-FLAG/tdTomato (WT) line proliferated normally in the presence of IAA, indicating that IAA is not toxic to parasites (Fig 2G). By contrast, the r*Tg*Tah18-mAID-HA, r*Tg*Dre2-mAID-HA, rHA-mAID-*Tg*Nar1, r*Tg*CIA1-mAID-HA, r*Tg*CIA2-mAID-HA, and r*Tg*MMS19-mAID-HA lines all exhibited a severe proliferation defect when cultured in the presence of IAA (Fig 2A–2F), with the phenotype upon the depletion of *Tg*Nar1 consistent with that reported in another recent study [20]. These data indicate that *Tg*Tah18, *Tg*Dre2, *Tg*Nar1, *Tg*CIA1, *Tg*CIA2 and *Tg*MMS19 are all critical for parasite proliferation.

Next, we wanted to explore whether the candidate proteins function in the CIA pathway. Given its curious dual localization to the mitochondrion and cytosol, we initially focused on the CTC protein *Tg*CIA1. Previous studies have found that impairment of the cytosolic FeS assembly pathway in *T. gondii* leads to a depletion in the abundances of some cytosolic and nuclear FeS cluster-containing proteins, including the ribosome recycling protein *Tg*ABCE1 [15,20–22]. We integrated a 3× cMyc tag at the 3′ end of the *Tg*ABCE1 open reading frame in the r*Tg*CIA1-mAID-HA line using CRISPR/Cas9 genome editing (S3A and S3B Fig). We cultured these parasites for 0, 24, or 48 hours on IAA, separated proteins by SDS-PAGE, and performed western blotting to detect the *Tg*ABCE1-cMyc protein. We found a small but significant decrease in *Tg*ABCE1-cMyc abundance after 24 h of IAA incubation and a strong ~75% decrease after 48 h (Fig 3A and 3B).

ABCE1 and other cytosolic FeS proteins are important for protein translation from cytosolic ribosomes. Previous studies have found that depletion of proteins that contribute to the CIA pathway result in a global impairment of protein synthesis in *T. gondii* [20,21]. We cultured WT and r*Tg*CIA1-mAID-HA parasites in the absence or presence of IAA for 24 h, then treated parasites with the protein synthesis inhibitor puromycin for 15 min. Puromycin becomes incorporated into nascent polypeptides during translation and can be detected by anti-puromycin antibodies, thus providing a measure for newly synthesized proteins. We separated proteins from puromycin-treated parasites using SDS-PAGE and performed western blotting. We observed a significant ~80% depletion in puromycin-labeled proteins in IAA-treated r*Tg*CIA1-mAID-HA parasites, whereas puromycin labeling was unchanged in IAA-treated WT parasites (Fig 3C and 3D). We next performed puromycin incorporation assays in r*Tg*Tah18-mAID-HA, r*Tg*Dre2-mAID-HA, rHA-mAID-*Tg*Nar1, r*Tg*CIA2-mAID-HA and r*Tg*MMS19-mAID-HA parasites cultured for 24 h in the absence or presence of IAA. We observed significantly less puromycin incorporation upon the depletion of *Tg*Tah18-mAID-HA, HA-mAID-*Tg*Nar1, *Tg*CIA2-mAID-HA and *Tg*MMS19-mAID-HA but not upon the depletion of *Tg*Dre2-mAID-HA (S4A and S4B Fig).

To test whether *Tg*Dre2 knockdown leads to the depletion of *Tg*ABCE1, we integrated a cMyc epitope tag into the *Tg*ABCE1 locus of r*Tg*Dre2-mAID-HA parasites (S3C Fig). We cultured these parasites for 0, 24, or 48 hours on IAA, separated proteins by SDS-PAGE, and performed western blotting to detect the *Tg*ABCE1-cMyc protein. We found a ~40%

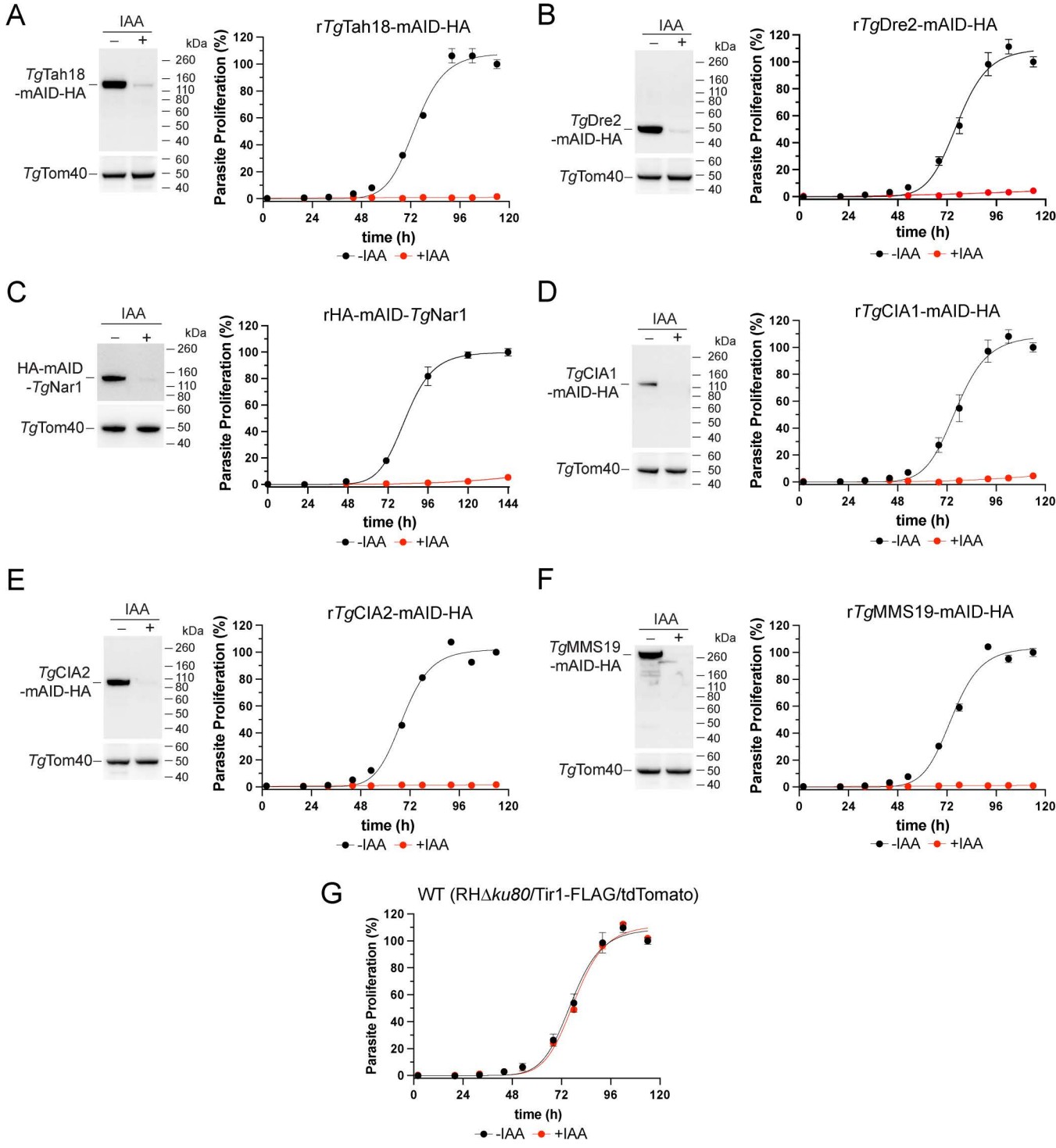

**Fig 2. Candidate CIA pathway proteins are important for parasite proliferation. (A–G)** Fluorescence proliferation assays and western blotting of **(A)** r*Tg*Tah18-mAID-HA, **(B)** r*Tg*Dre2-mAID-HA, **(C)** rHA-mAID-*Tg*Nar1, **(D)** r*Tg*CIA1-mAID-HA, **(E)** r*Tg*CIA2-mAID-HA, **(F)** r*Tg*MMS19-mAID-HA, and **(G)** WT (RHΔ*ku80*/Tir1-FLAG/tdTomato) parasites cultured in the absence (black) or presence (red) of IAA. Parasite proliferation is expressed as a percentage of the final fluorescence measurement in the -IAA condition for each line. Individual data points and error bars represent the mean ± SD of three technical replicates. Data are representative of three independent experiments. Error bars not visible are smaller than the symbol. The numerical data underlying this Figure can be found in S1 Data. For the western blotting, parasites were cultured in the absence (−) or presence (+) of IAA for 24 h, with

protein samples separated by SDS-PAGE then subjected to western blotting with anti-HA antibodies to detect the mAID-HA-tagged protein of interest and anti-*Tg*Tom40 antibodies as a loading control (**A–F**, left).

decrease in *Tg*ABCE1-cMyc abundance after 24 h of IAA incubation and a ~60% decrease after 48 h (S4C Fig). Loss of *Tg*Dre2, therefore, does impact the abundance of the cytosolic Fe–S protein *Tg*ABCE1, leaving open the possibility that *Tg*Dre2 has a role in cytosolic Fe–S synthesis. Taken together, our data indicate that the putative CIA pathway proteins of *T. gondii* are essential for parasite proliferation, and that most of the candidate CIA pathway proteins in *T. gondii* (with the possible exception of *Tg*Dre2) are critical for protein translation, a process that relies on cytosolic FeS proteins like ABCE1.

The CIA2 protein in animals and fungi contains a reactive cysteine residue that is thought to function in FeS cluster binding and facilitating the transfer of FeS clusters to client proteins [10,23,24]. This cysteine residue has been shown to be critical for CIA pathway function and cell survival in these eukaryotes [23,24]. Alignments of yeast and animal CIA2 proteins with *Tg*CIA2 revealed that the reactive cysteine residue is conserved as residue 524 in the *Tg*CIA2 protein (Figs 3E and S5). We hypothesized that if *Tg*CIA2 functions like its counterpart in other eukaryotes, mutating the $TgCIA2_{C524}$ residue would ablate CIA pathway function and, subsequently, lead to impaired parasite proliferation. We constitutively expressed either Ty1 epitope-tagged WT *Tg*CIA2 (constitutive (c)$TgCIA2_{WT}$-Ty1) or a Ty1-tagged *Tg*CIA2 mutant wherein the cysteine at residue 524 was mutated to Ala (c$TgCIA2_{C524A}$-Ty1) in r*Tg*CIA2-mAID-HA parasites. Western blotting revealed that both Ty1-tagged *Tg*CIA2 proteins were expressed at similar levels (Fig 3F) and fluorescence proliferation assays revealed that expression of c$TgCIA2_{WT}$-Ty1 fully rescued parasite proliferation upon knockdown of the *Tg*CIA2-mAID-HA protein (Fig 3G). By contrast, we observed minimal proliferation of r*Tg*CIA2-mAID-HA/c$TgCIA2_{C524A}$-Ty1 parasites cultured in the presence of IAA (Fig 3G). These data indicate that the proposed FeS cluster binding Cys-524 residue of *Tg*CIA2 is critical for its function, consistent with the hypothesis that the *Tg*CIA2 protein of *T. gondii* functions in a similar manner to the CIA2 protein of other eukaryotes in coordinating FeS cluster binding on the CTC [10,23,24].

### *Tg*CIA1, *Tg*CIA2, and *Tg*MMS19 are components of the same protein complex

We set out to explore the candidate CTC proteins of *T. gondii* in more detail. First, we asked whether the *Tg*CIA1, *Tg*CIA2 and *Tg*MMS19 proteins exist in a protein complex. We extracted proteins from *Tg*CIA1-HA, *Tg*CIA2-HA and *Tg*MMS19-HA expressing parasites, separated them by blue native (BN)-PAGE, and performed western blotting with anti-HA antibodies. All three proteins were present in a complex of slightly larger than 720 kDa (Fig 4A, black arrowhead). The *Tg*CIA1-HA protein, but not the others, was also observed in smaller protein complexes of ~400 and ~240 kDa (Fig 4A, red and blue arrowheads).

To test whether the other candidate CIA pathway proteins were present in protein complexes, we also undertook BN-PAGE western blotting on proteins extracted from the r*Tg*Tah18-mAID-HA, r*Tg*Dre2-mAID-HA and rHA-mAID-*Tg*Nar1 lines. We observed *Tg*Tah18-mAID-HA in a protein complex of ~330 kDa, but were unable to clearly detect the *Tg*Dre2-mAID-HA or HA-mAID-*Tg*Nar1 proteins in complexes (S6A Fig). These data suggest that *Tg*Tah18, *Tg*Dre2, and *Tg*Nar1 are not components of the >720 kDa CTC.

If the *Tg*CIA1, *Tg*CIA2, and *Tg*MMS19 proteins exist in the same complex, we reasoned that depleting one would result in alterations to the protein complex in which the other two proteins reside. We initially introduced Ty1-epitope tags into the native *Tg*CIA2 and *Tg*MMS19 loci of the r*Tg*CIA1-mAID-HA line, but were subsequently unable to detect robust *Tg*CIA2-Ty1 and *Tg*MMS19-Ty1 expression, possibly because the antibody we have to the Ty1 epitope is not of sufficiently high affinity. Since we knew we could detect the HA-tagged versions of these proteins, we generated a new IAA-regulatable *Tg*CIA1-expressing line by fusing a mAID-cMyc tag into the 3′ region of the *Tg*CIA1 open reading frame (S7A Fig). We then introduced a HA tag into the 3′ region of the *Tg*CIA2 or *Tg*MMS19 open reading frames of this parasite

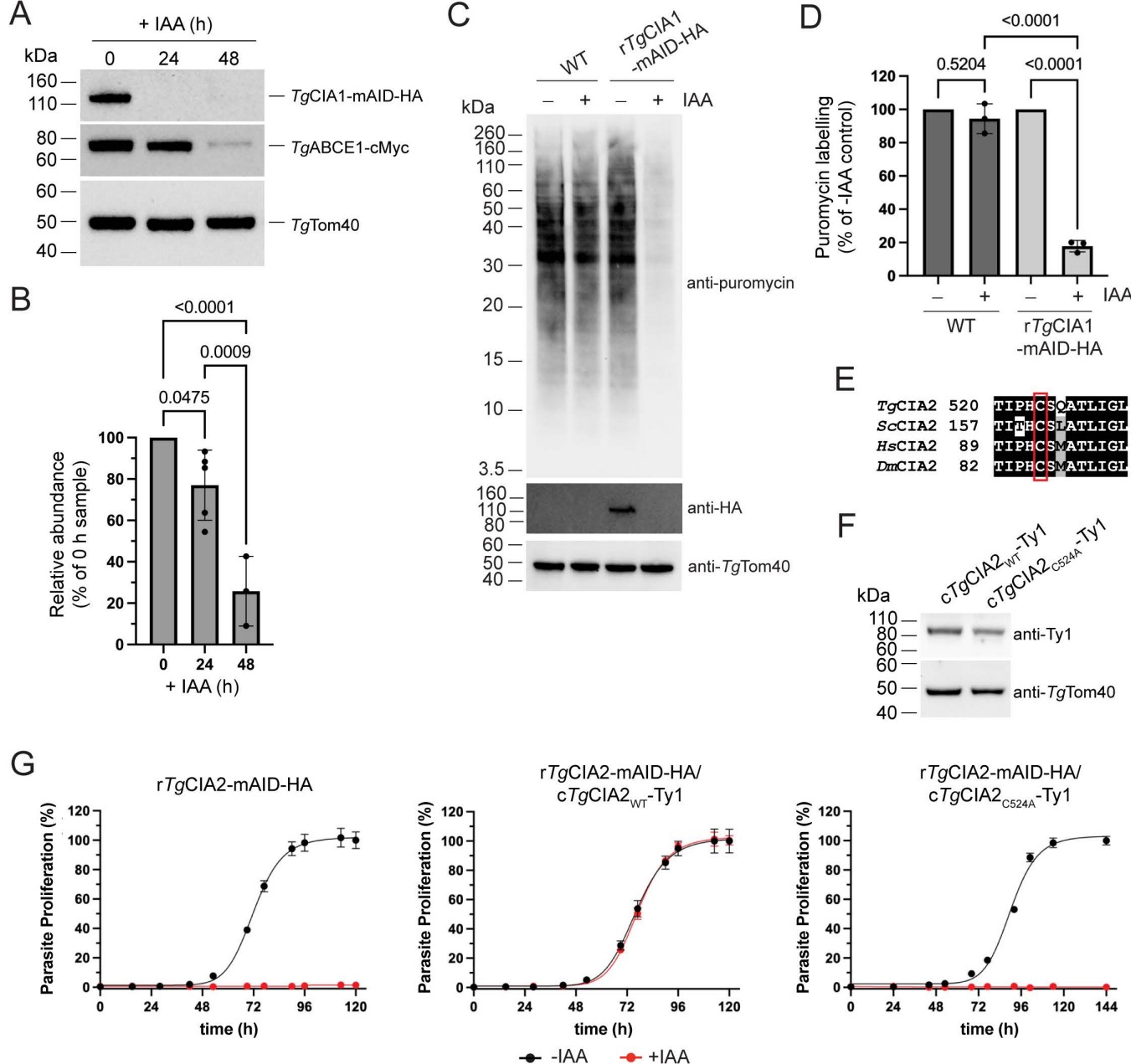

**Fig 3. Characterizing the effects of depleting candidate CIA pathway proteins in *Toxoplasma gondii* parasites. (A)** Western blots of proteins extracted from r*Tg*CIA1-mAID-HA/*Tg*ABCE1-cMyc parasites cultured for 0, 24 or 48 h in IAA and separated by SDS-PAGE. Samples were probed with anti-HA, anti-cMyc, and anti-*Tg*Tom40 antibodies. **(B)** Relative abundance of the *Tg*ABCE1-cMyc protein in the western blot was determined as a percentage of the 0 h control, with abundances normalized using the *Tg*Tom40 loading control. Data points represent the mean ± SD of five independent experiments for the 0 and 24 h conditions and three independent experiments for the 48 h condition. Data were analyzed using a one-way ANOVA followed by Tukey's multiple comparisons test, with *p* values shown. **(C)** Western blots measuring the incorporation of puromycin into proteins from WT (RHΔ*ku80*-Tir1-FLAG/tdTomato) or r*Tg*CIA1-mAID-HA parasites cultured in the absence (−) or presence (+) of IAA for 24 h, separated by SDS-PAGE and probed with anti-puromycin, anti-HA, and anti-*Tg*Tom40 antibodies. **(D)** Relative abundance of puromycin incorporation was determined as a percentage of the -IAA control for each parasite line, with abundances normalized using the *Tg*Tom40 loading control. Data points represent the mean ± SD of three independent experiments. Data were analyzed using a one-way ANOVA followed by Tukey's multiple comparisons test, with relevant *p* values shown. **(E)** Multiple sequence alignment of a region of the *T. gondii* CIA2 protein (*Tg*CIA2) with CIA2 homologs in *Saccharomyces cerevisiae* (*Sc*CIA2), *Homo sapiens* (*Hs*CIA2), and *Drosophila melanogaster* (*Dm*CIA2). The cysteine residue proposed to function in FeS cluster binding is highlighted by a red box. **(F)** Western blot of proteins extracted from r*Tg*CIA2-mAID-HA/c*Tg*CIA2_WT-Ty1 or r*Tg*CIA2-mAID-HA/c*Tg*CIA2_C524A-Ty1 parasites, separated

by SDS-PAGE and probed with anti-Ty1 antibodies to detect the c*Tg*CIA2$_{WT}$-Ty1 and c*Tg*CIA2$_{C524A}$-Ty1 proteins, and anti-*Tg*Tom40 antibodies as a loading control. **(G)** Fluorescence proliferation assays of r*Tg*CIA2-mAID-HA, r*Tg*CIA2-mAID-HA/c*Tg*CIA2$_{WT}$-Ty1, or r*Tg*CIA2-mAID-HA/c*Tg*CIA2$_{C524A}$-Ty1 parasites cultured in the absence (black) or presence (red) of IAA. Parasite proliferation is expressed as a percentage of the -IAA condition on the final day of the experiment for each parasite line, with values depicting the mean±SD of three technical replicates. Error bars not visible are smaller than the symbol. Data are representative of three independent experiments. The numerical data underlying this Figure can be found in S1 Data.

line (S7B and S7C Fig). We cultured the resulting r*Tg*CIA1-mAID-cMyc/*Tg*CIA2-HA and r*Tg*CIA1-mAID-cMyc/*Tg*MMS19-HA parasites in the absence or presence of IAA for 24 h, separated proteins by SDS-PAGE, and performed western blotting. We observed robust depletion of the *Tg*CIA1-mAID-cMyc protein upon the addition of IAA (Fig 4B). We also a observed an ~80% depletion of the *Tg*CIA2-HA protein, and a small but significant ~20% depletion of the *Tg*MMS19-HA protein (Fig 4B and 4C). To determine the timing of the observed *Tg*CIA2-HA depletion upon *Tg*CIA1 knockdown, we cultured r*Tg*CIA1-mAID-cMyc/*Tg*CIA2-HA parasites in IAA for a range of times between 0 and 12 h. We observed a depletion of the *Tg*CIA1-mAID-cMyc protein as soon as 3 h after IAA addition, and a concomitant and significant decrease in the abundance of the *Tg*CIA2-HA protein (S6B Fig). These data indicate that *Tg*CIA1 knockdown impacts the stability of the *Tg*CIA2 and, to a lesser extent, the *Tg*MMS19 proteins.

We next extracted proteins from r*Tg*CIA1-mAID-cMyc/*Tg*CIA2-HA and r*Tg*CIA1-mAID-cMyc/*Tg*MMS19-HA parasites cultured in the absence or presence of IAA for 24 h, separated proteins by BN-PAGE, and performed western blotting. *Tg*CIA1-mAID-cMyc knockdown resulted in a depletion and reduced molecular mass in both the *Tg*CIA2-HA- and *Tg*MMS19-HA-containing protein complexes (Fig 4D), with the depletion of the *Tg*CIA2-HA complex occurring concomitantly with *Tg*CIA1-mAID-cMyc knockdown (S6C Fig).

We next undertook the reverse experiment, asking whether depletion of *Tg*CIA2 or *Tg*MMS19 resulted in changes in the abundance of *Tg*CIA1 and impairment of the *Tg*CIA1-containing protein complexes. We integrated a spaghetti monster-fluorescent protein-cMyc (smFP-cMyc) tag into the *Tg*CIA1 locus of r*Tg*CIA2-mAID-HA and r*Tg*MMS19-mAID-HA parasites (S7D Fig). We cultured the r*Tg*CIA2-mAID-HA/*Tg*CIA1-smFP-cMyc and r*Tg*MMS19-mAID-HA/*Tg*CIA1-smFP-cMyc lines in the absence or presence of IAA for 24 h and performed both SDS-PAGE and BN-PAGE western blotting. We observed no changes to *Tg*CIA1-smFP-cMyc protein abundance upon the knockdown of either *Tg*CIA2-mAID-HA or *Tg*MMS19-mAID-HA (Fig 4E). However, we observed a complete loss of the >720 kDa *Tg*CIA1-smFP-cMyc-containing complex upon the knockdown of both *Tg*CIA2-mAID-HA and *Tg*MMS19-mAID-HA (Fig 4F, black arrowhead). The two smaller *Tg*CIA1-containing complexes were unaffected by *Tg*CIA2 or *Tg*MMS19 depletion (Fig 4F, red and blue arrowheads).

As a direct test for whether *Tg*CIA1 and *Tg*CIA2 interact, we performed co-immunoprecipitation experiments on the r*Tg*CIA2-mAID-HA/*Tg*CIA1-smFP-cMyc parasite line. Immunoprecipitation of *Tg*CIA1-smFP-cMyc with anti-cMyc beads co-purified *Tg*CIA2-mAID-HA but not the mitochondrial outer membrane protein *Tg*Tom40, and immunoprecipitation of *Tg*CIA2-mAID-HA with anti-HA beads co-purified *Tg*CIA1-smFP-cMyc but not *Tg*Tom40 (S6D Fig), providing further evidence that *Tg*CIA1 and *Tg*CIA2 are components of the same protein complex.

Taken together, our data are consistent with *Tg*CIA1, *Tg*CIA2, and *Tg*MMS19 existing in the same complex of >720 kDa. The CTC of animals exists as a heteromeric complex consisting of two copies of each of the three proteins [10]. The combined mass of a similarly arranged complex in *T. gondii* would be ~850 kDa, which is conceivably the mass we observe for the *Tg*CIA1, *Tg*CIA2, and *Tg*MMS19 complexes in BN-PAGE. Our data also indicate that *Tg*CIA1 exists in smaller protein complexes that do not include *Tg*CIA2 or *Tg*MMS19.

## A novel loop in the *Tg*CIA1 protein facilitates the dual cytosolic and mitochondrial localization of the CIA targeting complex in *T. gondii*

The candidate CTC proteins of *T. gondii* exhibit an interesting dual localization to the mitochondrion and cytosol that is suggestive of a dynamic role for this complex in parasite biology. We set out to characterize the mitochondrial targeting

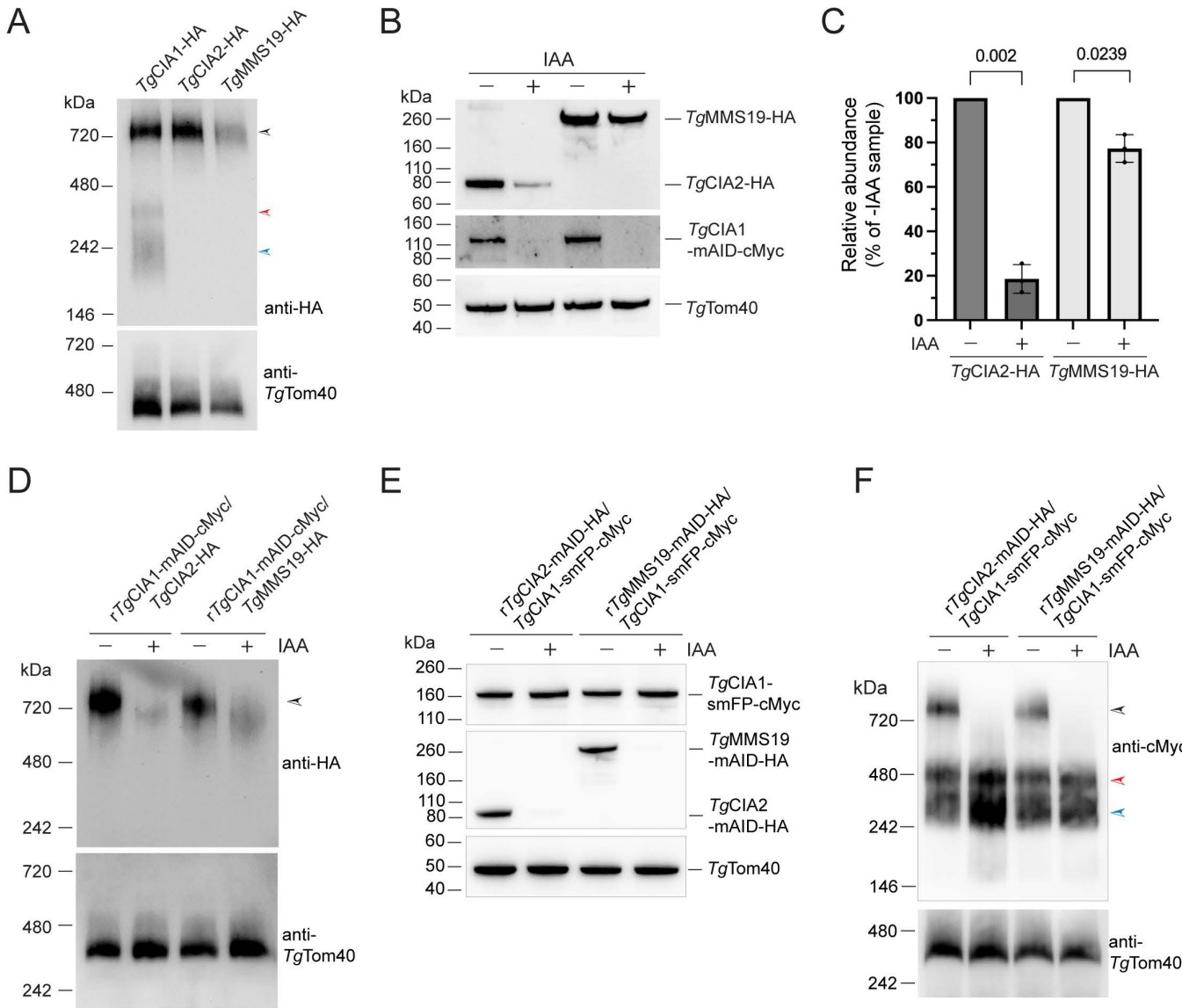

**Fig 4. Characterizing the protein composition of the CIA Targeting Complex (CTC) in *Toxoplasma gondii* parasites. (A)** Western blot of proteins extracted from *Tg*CIA1-HA, *Tg*CIA2-HA and *Tg*MMS19-HA expressing parasites, separated by BN-PAGE and probed with anti-HA antibodies or anti-*Tg*Tom40 as a loading control. **(B, D)** Western blots of proteins extracted from r*Tg*CIA1-mAID-cMyc/*Tg*CIA2-HA and r*Tg*CIA1-mAID-cMyc/ *Tg*MMS19-HA parasites cultured in the absence (-) or presence (+) of IAA for 24 h, separated by either SDS-PAGE **(B)** or BN-PAGE **(D)** and probed with anti-HA, anti-cMyc or anti-*Tg*Tom40 antibodies. Data are representative of three independent experiments. **(C)** Quantification of the western blots depicted in **(B)**, depicting relative abundance of the *Tg*CIA2-HA or *Tg*MMS19-HA proteins determined as a percentage of the -IAA control, with abundances normalized using the *Tg*Tom40 loading control. Data points represent the mean ± SD of three independent experiments. Data were analyzed using a paired *t* test with *p* values shown. The numerical data underlying this Figure can be found in S1 Data. **(E, F)** Western blot of proteins extracted from r*Tg*CIA2-mAID-HA/*Tg*CIA1-smFP-cMyc and r*Tg*MMS19-mAID-HA/*Tg*CIA1-smFP-cMyc parasites, cultured for 24 h in the absence (−) or presence (+) of IAA, separated by either **(E)** SDS-PAGE or **(F)** BN-PAGE and probed with anti-cMyc, anti-HA, or anti-*Tg*Tom40 antibodies. Data are representative of three independent experiments. In the BN-PAGE western blots (A, D, and F), the black arrowheads indicate the >720 kDa candidate CIA Targeting Complex; red and blue arrowheads indicate the lower mass complexes containing the *Tg*CIA1 protein.

of the CTC in more detail. We first asked whether *Tg*CIA1 was required for the mitochondrial targeting of *Tg*CIA2 and *Tg*MMS19. We performed immunofluorescence assays on r*Tg*CIA1-mAID-cMyc/*Tg*CIA2-HA or r*Tg*CIA1-mAID-cMyc/*Tg*MMS19-HA parasites cultured for 3 h in the absence or presence of IAA, probing for the HA-tagged *Tg*CIA2 and *Tg*MMS19 proteins. As expected, both *Tg*CIA2-HA and *Tg*MMS19-HA exhibited dual mitochondrial and cytosolic localization in parasites cultured in the absence of IAA (Fig 5A and 5B). Notably, however, the mitochondrial localization of both *Tg*CIA2-HA and *Tg*MMS19-HA was significantly reduced upon *Tg*CIA1-mAID-cMyc depletion (Fig 5A and 5B). This indicates that both *Tg*CIA2 and *Tg*MMS19 depend on *Tg*CIA1 for their mitochondrial targeting.

We next asked whether the mitochondrial localization of *Tg*CIA1 is dependent on either *Tg*CIA2 or *Tg*MMS19. We cultured r*Tg*CIA2-mAID-HA/*Tg*CIA1-smFP-cMyc or r*Tg*MMS19-mAID-HA/*Tg*CIA1-smFP-cMyc parasites in the absence or presence of IAA for 24 h and performed immunofluorescence assays. Neither the depletion of *Tg*CIA2-mAID-HA nor *Tg*MMS19-mAID-HA significantly altered the localization of *Tg*CIA1-smFP-cMyc (Fig 5C and 5D), indicating that the mitochondrial localization of *Tg*CIA1 is not dependent on either *Tg*CIA2 or *Tg*MMS19. Taken together, these data indicate that targeting of the CTC to the mitochondrion of *T. gondii* is mediated by *Tg*CIA1.

Next, we asked whether *Tg*CIA1 targeting to the mitochondrion is the result of interactions with other mitochondrially-localized components of the CIA pathway. We showed previously that *Tg*NBP35, the proposed scaffold of the CIA pathway, is anchored to the mitochondrial outer membrane courtesy of an N-terminal TMD [15]. We cultured r*Tg*NBP35-cMyc/*Tg*CIA1-HA parasites in the absence or presence of ATc for 2 days to deplete *Tg*NBP35-cMyc abundance, then performed immunofluorescence assays with anti-HA antibodies to determine *Tg*CIA1-HA localization. We found that *Tg*CIA1-HA continued to localize to the mitochondrion following *Tg*NBP35-cMyc depletion (S8A Fig). We next tested whether *Tg*Tah18, another mitochondrially-localized component of the CIA pathway in *T. gondii* (Fig 1G), is important for the mitochondrial localization of *Tg*CIA1. We integrated a smFP-cMyc tag into the *Tg*CIA1 locus of r*Tg*Tah18-mAID-HA parasites (S8B Fig). We cultured r*Tg*Tah18-mAID-HA/*Tg*CIA1-smFP-cMyc parasites in the absence or presence of IAA for 24 h and performed immunofluorescence assays. We found that the localization of *Tg*CIA1-smFP-cMyc was unchanged upon *Tg*Tah18-mAID-HA depletion (S8C Fig). Taken together, these data indicate that the mitochondrial targeting of *Tg*CIA1 is independent of *Tg*NBP35 and *Tg*Tah18.

A recent study found that Nar1 homologs from other eukaryotes harbor a C-terminal amino acid motif that facilitates Nar1 interaction with the CTC [25]. This motif, which includes a C-terminal tryptophan residue, appears to be conserved in *Tg*Nar1 (S8D Fig), which could explain why we were unable to C-terminally tag this protein. To test whether mitochondrially-localized *Tg*Nar1 facilitates the mitochondrial localization of *Tg*CIA1, we introduced a smFP-cMyc tag into the *Tg*CIA1 locus of rHA-mAID-*Tg*Nar1 parasites (S8B Fig). We cultured the resulting rHA-mAID-*Tg*Nar1/*Tg*CIA1-smFP-cMyc parasites in the absence or presence of IAA for 24 h and performed both SDS-PAGE and BN-PAGE western blotting. Depleting HA-mAID-*Tg*Nar1 did not affect the abundance of the *Tg*CIA1-smFP-cMyc protein or the formation of the *Tg*CIA1-smFP-cMyc-containing protein complexes (S8E and S8F Fig). We were also unable to detect *Tg*CIA1-smFP-cMyc protein upon immunoprecipitation of HA-mAID-*Tg*Nar1 protein (S8G Fig), suggesting that these proteins do not stably interact. This is consistent with our BN-PAGE data that *Tg*Nar1 is not part of the CTC (S6A Fig), although our data do not rule out that transient interactions occur between these proteins. Finally, we cultured rHA-mAID-*Tg*Nar1/*Tg*CIA1-smFP-cMyc parasites in the absence or presence of IAA for 24 h and performed immunofluorescence assays to detect the *Tg*CIA1-smFP-cMyc protein. We found that the dual mitochondrial/cytosolic localization of *Tg*CIA1-smFP-cMyc was unchanged upon HA-mAID-*Tg*Nar1 depletion (S8H Fig). Taken together, these data indicate that the mitochondrial localization of *Tg*CIA1 is not due to interactions with mitochondrially localized *Tg*Nar1.

We looked to the AlphaFold2 predicted structure of *Tg*CIA1 for clues into what could be mediating the targeting of *Tg*CIA1 (and by extension, the CTC) to the mitochondrion. CIA1 belongs to the WD40 protein family, which commonly mediate protein-protein interactions [26,27]. The CIA1 proteins from yeast and animals contain characteristic WD40 repeat domains folded into a 7-bladed β-propeller arranged around a central axis (Fig 6A; [10,28]). Each WD40 domain

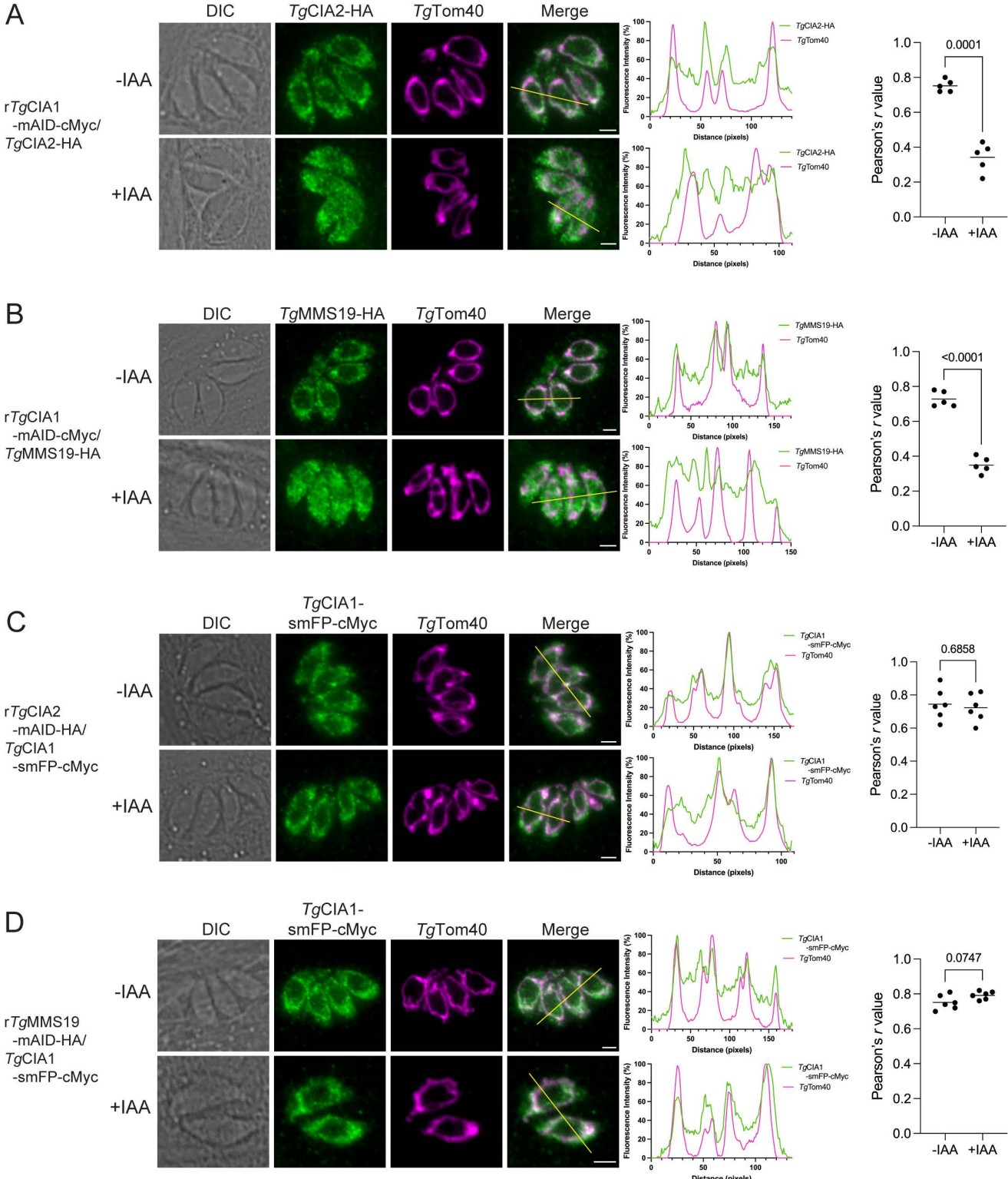

**Fig 5. Localization of the CTC to the mitochondrion of *Toxoplasma gondii* is mediated by *Tg*CIA1. (A–D)** Immunofluorescence assays of **(A)** r*Tg*CIA1-mAID-cMyc/*Tg*CIA2-HA, **(B)** r*Tg*CIA1-mAID-cMyc/*Tg*MMS19-HA, **(C)** r*Tg*CIA2-mAID-HA/*Tg*CIA1-smFP-cMyc, and **(D)** r*Tg*MMS19-mAID-HA/ *Tg*CIA1-smFP-cMyc parasites, cultured in the absence (top) or presence (bottom) of IAA for either 3 h **(A, B)** or 24 h (C, D). Samples were probed with anti-HA **(A, B)** or anti-cMyc **(C, D)** antibodies to detect the protein-of-interest (green) and anti-*Tg*Tom40 antibodies to detect the mitochondrion

(magenta). Scale bars are 2 µm. DIC, differential interference contrast. Middle, corresponding fluorescence profiles depicting the intensity of anti-HA or anti-cMyc labeling (green) and the anti-*Tg*Tom40 labeling (magenta) along the yellow line on merged images. Right, the correlation between the protein of interest and *Tg*Tom40 was quantified using the Pearson correlation coefficient (*r*) and analyzed using an unpaired *t* test, with the *p* values shown. The numerical data underlying this Figure can be found in S1 Data.

(or "blade") is comprised of four antiparallel β-strands (denoted A–D in Fig 6A, pink) with small loops of 5–12 amino acids connecting the C and D strands (CD loops; Fig 6A; [10,28]). In the AlphaFold2 predicted structure of *Tg*CIA1, the 7-bladed β-propeller is structurally conserved. However, the *Tg*CIA1 protein contains three large insertions compared to the yeast and animal proteins (Figs 6A and S9; [10,28]). These insertions are located in the CD loops of the first, third, and fifth β-propeller blades, and we termed these the CD1, CD3, and CD5 loops, respectively (Fig 6A). The CD1 loop is 122 amino acids long in *Tg*CIA1, the CD3 loop is 82 amino acids long, and the CD5 loop is 289 amino acids (S9 Fig). Extended CD loops are also present in the CIA1 homologs of other myzozoans (S9 Fig), a eukaryotic taxon that includes apicomplexans and their closest free-living relatives such as chrompodellids and dinozoans [13,14]. This is most apparent in the CD5 loop of myzozoan CIA1 proteins, which ranges in length from 45 amino acids in the oyster parasite *P. marinus* to 313 amino acids in the malaria-causing parasite *P. falciparum* (S9 Fig). This is in contrast to the much shorter, six-amino acid-long CD5 loop of CIA1 in the ciliate *Paramecium tetraurelia*, a sister taxon to the myzozoans. The extended CD loops in the *Tg*CIA1 structure are predicted with poor confidence by AlphaFold2, and for the most part lack clear structural elements (Fig 6A).

Notably, the CD loops are located on the opposite side of the CIA1 protein from that which interacts with CIA2 [10]. This places the CD loops in position to be involved in interactions external to the CTC. We therefore tested whether the extended CD1, CD3, or CD5 loops of *Tg*CIA1 could have a role in mitochondrial targeting. We generated *Tg*CIA1 trans-genes in which the endogenous CD1, CD3, or CD5 loops of *Tg*CIA1 were substituted for the equivalent, but substantially shorter, loops from the cytosolic CIA1 protein of yeast. We fused the resulting transgenes to a Ty1 epitope tag to enable their detection, and overexpressed them from the constitutive α-tubulin promoter in the r*Tg*CIA1-mAID-HA line. We termed the resulting proteins c*Tg*CIA1$_{ScCD1}$-Ty1, c*Tg*CIA1$_{ScCD3}$-Ty1, and c*Tg*CIA1$_{ScCD5}$-Ty1, respectively, and also included a c*Tg*CIA1$_{WT}$-Ty1 control in our analyses. SDS-PAGE western blot analysis confirmed that each protein was expressed in *T. gondii* and were of the expected masses (Fig 6B). We observed some differences in expression levels between the constitutively-expressed proteins, with the c*Tg*CIA1$_{WT}$-Ty1 and c*Tg*CIA1$_{ScCD1}$-Ty1 proteins more abundant than the c*Tg*CIA1$_{ScCD3}$-Ty1 and c*Tg*CIA1$_{ScCD5}$-Ty1 proteins (S10A Fig). To test the localization of the modified proteins, we performed immunofluorescence assays. As expected, the c*Tg*CIA1$_{WT}$-Ty1 protein exhibited dual localization to the cytosol and mito-chondrion (Fig 6C). Both c*Tg*CIA1$_{ScCD1}$-Ty1 and c*Tg*CIA1$_{ScCD3}$-Ty1 proteins also localized dually to the mitochondrion and cytosol (Fig 6C). This infers that the native CD1 and CD3 loops of *Tg*CIA1 are not important for mitochondrial targeting, although we observed that r*Tg*CIA1-mAID-HA/c*Tg*CIA1$_{ScCD1}$-Ty1 parasites exhibited aberrant mitochondrial morphology, with ~75% of parasites containing "tadpole"-like mitochondria instead of the typical "lasso" and 'branched' structures that mitochondria adopt in intracellular parasites (S10B Fig; [29]). In contrast to the other CD loop mutants, the mitochondrial localization of the c*Tg*CIA1$_{ScCD5}$-Ty1 protein was significantly reduced, with the protein localized predominantly to the cytosol (Fig 6C and 6D). This indicates that the extended CD5 loop of *Tg*CIA1 is necessary for mitochondrial targeting.

We next explored the importance of the extended CD loops for *Tg*CIA1 function. We first performed BN-PAGE west-ern blotting. Each of the CD loop mutants associated with a complex of ~720 kDa, which likely represents the CTC (Fig 6E; black arrowhead). This implies that the CD1, CD3, and CD5 loops are not required for *Tg*CIA1 to assemble into the CTC. The c*Tg*CIA1$_{WT}$-Ty1, c*Tg*CIA1$_{ScCD1}$-Ty1, and c*Tg*CIA1$_{ScCD3}$-Ty1 proteins were all found in the smaller TgCIA1 com-plexes (Fig 6E; red and blue arrowheads), similar in mass to those we had observed previously with the natively tagged *Tg*CIA1-HA protein (Fig 4A). The smaller *Tg*CIA1 complexes in the c*Tg*CIA1$_{WT}$-Ty1 and c*Tg*CIA1$_{ScCD1}$-Ty1 lines were, in proportion to the ~720 kDa complex, more abundant than for the natively tagged *Tg*CIA1-HA protein (compare Figs 4A to

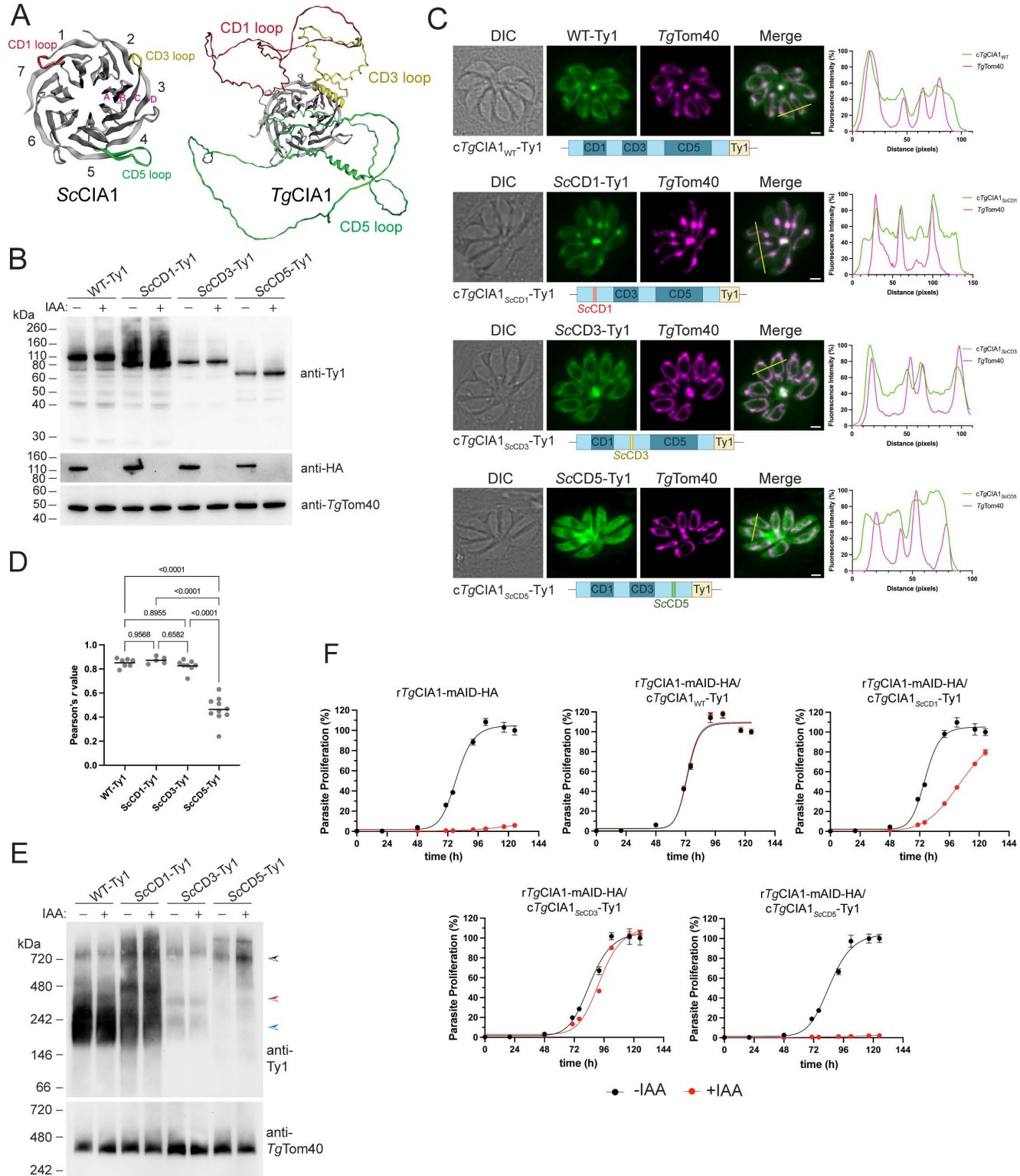

**Fig 6. An elongated CD5 loop is necessary for targeting *Tg*CIA1 to the mitochondrion of *Toxoplasma gondii*. (A)** Structure of the *Saccharomyces cerevisiae* protein *Sc*CIA1 (PDB: 2HES) and the predicted AlphaFold2 structure of *Tg*CIA1(UniProt ID: A0A125YRQ0), with the A, B, C, and D strands of the third propellor blade in the *Sc*CIA1 protein labeled in pink. The loops connecting the C and D strands of blade 1 (CD1 loop; red),

blade 3 (CD3 loop, yellow), and blade 5 (CD5 loop; green) in each structure were colored using EzMol software [63]. **(B, E)** Western blots of proteins extracted from r*Tg*CIA1-mAID-HA/c*Tg*CIA1$_{WT}$-Ty1 (WT-Ty1), r*Tg*CIA1-mAID-HA/c*Tg*CIA1$_{ScCD1}$-Ty1 (*Sc*CD1-Ty1), r*Tg*CIA1-mAID-HA/c*Tg*CIA1$_{ScCD3}$-Ty1 (*Sc*CD3-Ty1), and r*Tg*CIA1-mAID-HA/c*Tg*CIA1$_{ScCD5}$-Ty1 (*Sc*CD5-Ty1) parasites cultured in the absence or presence of IAA for 24 h, separated by **(B)** SDS-PAGE or **(E)** BN-PAGE, and probed with anti-Ty1, anti-HA, or anti-*Tg*Tom40 antibodies. The masses of the predicted CTC (black arrowhead) and smaller *Tg*CIA1-containing complexes (red and blue arrowheads) are depicted in the BN-PAGE data. **(C)** Immunofluorescence assays of r*Tg*CIA1-mAID-HA/c*Tg*CIA1$_{WT}$-Ty1 (WT-Ty1), r*Tg*CIA1-mAID-HA/c*Tg*CIA1$_{ScCD1}$-Ty1 (*Sc*CD1-Ty1), r*Tg*CIA1-mAID-HA/c*Tg*CIA1$_{ScCD3}$-Ty1 (*Sc*CD3-Ty1), and r*Tg*CIA1-mAID-HA/c*Tg*CIA1$_{ScCD5}$-Ty1 (*Sc*CD5-Ty1) parasites, probed with anti-Ty1 (green) and anti-*Tg*Tom40 (magenta) antibodies. Scale bars are 2 µm. DIC, differential inference contrast. Right, corresponding fluorescence profiles depicting the intensity of anti-Ty1 (green) and anti-*Tg*Tom40 (magenta) labeling along the yellow line on merged images. **(D)** The correlation between Ty1-tagged proteins and *Tg*Tom40 was quantified using the Pearson correlation coefficient (*r*). The data were analyzed using a one-way ANOVA followed by Tukey's multiple comparisons test with *p* values shown. **(F)** Fluorescence proliferation assays of r*Tg*CIA1-mAID-HA, r*Tg*CIA1-mAID-HA/c*Tg*CIA1$_{WT}$-Ty1, r*Tg*CIA1-mAID-HA/c*Tg*CIA1$_{ScCD1}$-Ty1, r*Tg*CIA1-mAID-HA/c*Tg*CIA1$_{ScCD3}$-Ty1, and r*Tg*CIA1-mAID-HA/c*Tg*CIA1$_{ScCD5}$-Ty1 parasites cultured in the absence (black) or presence (red) of IAA. Parasite proliferation is expressed as a percentage of the final fluorescence measurement in the -IAA condition for each line. Individual data points and error bars represent the mean ± SD of three technical replicates. Error bars not visible are smaller than the symbol. Data are representative of three independent experiments. The numerical data underlying this Figure can be found in S1 Data.

6E), possibly an artifact of protein overexpression from the non-native α-tubulin promoter. Curiously, the c*Tg*CIA1$_{ScCD5}$-Ty1 protein did not appear to be present in the smaller mass complexes (Fig 6E), suggesting a role for the CD5 loop in assembly of these complexes.

Next, we tested whether the CD1, CD3, or CD5 loops of *Tg*CIA1 are important for parasite proliferation. We conducted fluorescence proliferation and plaque assays on r*Tg*CIA1-mAID-HA parasites constitutively expressing WT *Tg*CIA1 or the CD loop mutants in the absence or presence of IAA. As expected, the severe proliferation defect observed upon *Tg*CIA1-mAID-HA depletion was rescued by constitutive expression of c*Tg*CIA1$_{WT}$-Ty1 (Figs 6F and S10C). c*Tg*CIA1$_{ScCD1}$-Ty1-expressing parasites exhibited a moderate proliferation defect upon *Tg*CIA1-mAID-HA depletion, resulting in fewer (rather than smaller) plaques (Figs 6F and S10C). This suggests a possible role for the CD1 loop in processes that affect the viability of extracellular parasites or the ability of parasites to invade host cells. Parasites expressing the c*Tg*CIA1$_{ScCD3}$-Ty1 protein exhibited only minor defects in proliferation when the *Tg*CIA1-mAID-HA protein was depleted (Figs 6F and S10C). Notably, c*Tg*CIA1$_{ScCD5}$-Ty1-expressing parasites exhibited a severe impairment of parasite proliferation when *Tg*CIA1-mAID-HA was depleted, indistinguishable from the proliferation defect observed in non-complemented r*Tg*CIA1-mAID-HA parasites (Figs 6F and S10C). This indicates that the CD5 loop is critical for *Tg*CIA1 protein function.

Having demonstrated that the CD5 loop of *Tg*CIA1 is necessary for mitochondrial targeting, we wondered whether the CD5 loop alone could mediate mitochondrial protein targeting. We inserted the CD5 loop of *Tg*CIA1 into a green fluorescent protein (GFP)-Ty1 reporter at an internal site in GFP shown previously to tolerate insertions (between the eighth and ninth β-strands of GFP; [30]). We expressed GFP$_{TgCD5}$-Ty1 in *T. gondii* parasites and attempted to select parasites stably expressing the transgene. We found that, following selection, very few parasites expressed the GFP$_{TgCD5}$-Ty1 protein. In those that did, the GFP$_{TgCD5}$-Ty1 protein exhibited a dual localization to both the mitochondrion and cytosol (S11 Fig). However, we noticed that the mitochondrial morphology in these parasites appeared aberrant, suggesting a potential toxic effect of GFP$_{TgCD5}$-Ty1 overexpression that complicates our interpretation of the data.

We next generated a transgene encoding the structurally characterized *Drosophila melanogaster* CIA1 protein (*Dm*CIA1; [10]) in which we replaced the native CD5 loop of *Dm*CIA1 with the *Tg*CIA1 CD5 loop. We expressed c*Dm*CIA1$_{TgCD5}$-Ty1 or a corresponding c*Dm*CIA1$_{WT}$-Ty1 protein in r*Tg*CIA1-mAID-HA parasites and performed immunofluorescence assays to determine protein localization. As expected, c*Dm*CIA1$_{WT}$-Ty1 localized in the cytosol and did not overlap with the mitochondrion (Fig 7A). By contrast, the c*Dm*CIA1$_{TgCD5}$-Ty1 protein co-localized with the mitochondrial marker (Fig 7A). Taken together, these data indicate that the CD5 loop of *Tg*CIA1 is sufficient to mediate mitochondrial localization.

We next asked whether the position of the CD5 loop in the *Tg*CIA1 protein was important for mitochondrial localization or protein function. We engineered a *Tg*CIA1 construct in which the amino acid sequence encoding the CD5 loop of *Tg*CIA1 was located in the CD1 loop of the protein, with the native CD5 loop replaced by the equivalent CD5 loop of yeast

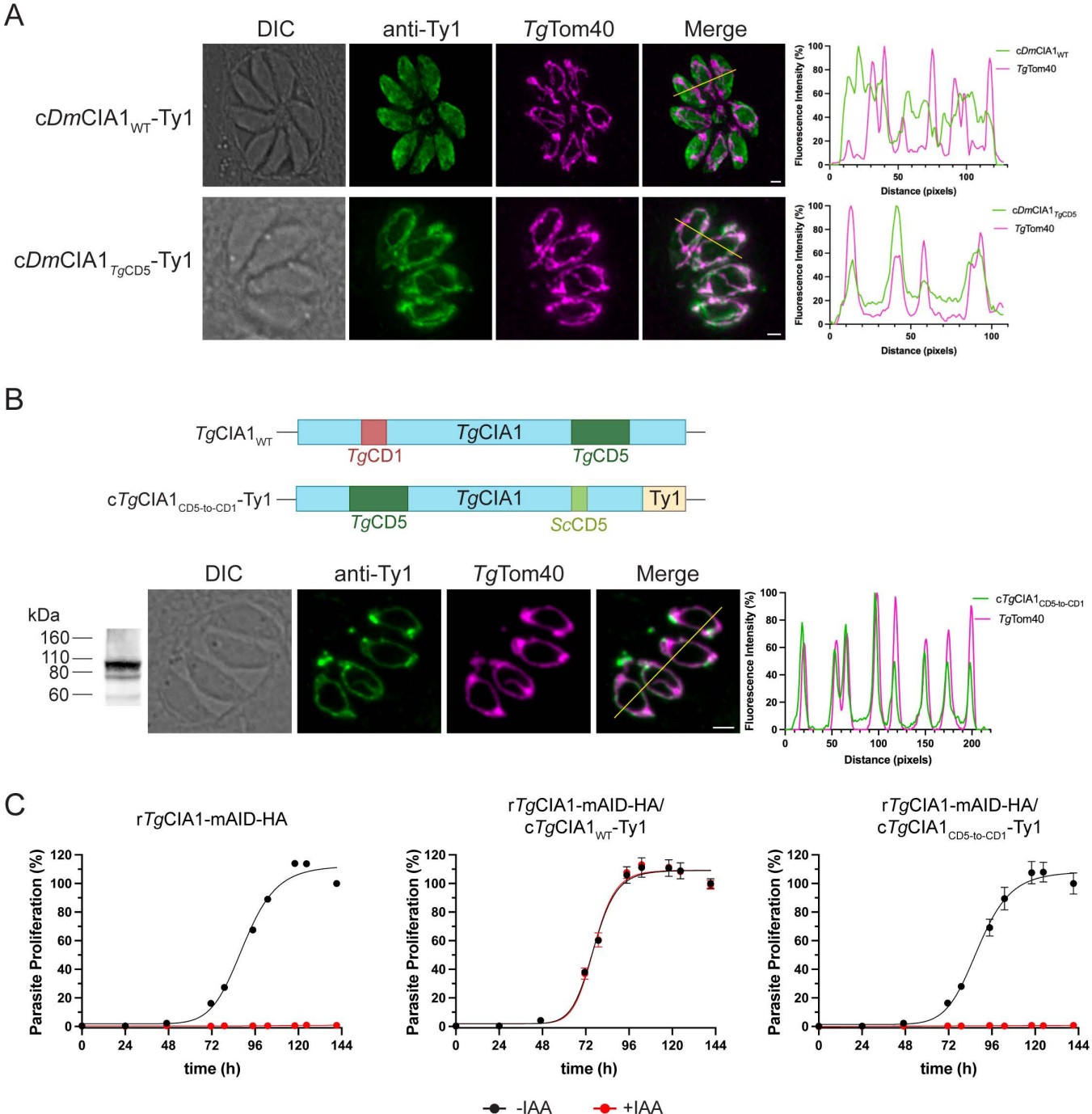

**Fig 7. The CD5 loop of *Tg*CIA1 is sufficient for mitochondrial targeting, with the positioning of the loop important for *Tg*CIA1 function. (A)** Immunofluorescence assays of parasites constitutively expressing c*Dm*CIA1$_{WT}$-Ty1 (top) or c*Dm*CIA1$_{TgCD5}$-Ty1, (bottom) probed with anti-Ty1 (green) and anti-*Tg*Tom40 (magenta; mitochondrion) antibodies. Scale bars are 2 μm. DIC, differential interference contrast. Right, corresponding fluorescence plots depicting the intensity of anti-Ty1 (green) and anti-*Tg*Tom40 (magenta) labeling along the yellow line in merged images. **(B)** Constitutive expression of a c*Tg*CIA1$_{CD5-to-CD1}$-Ty1 protein variant in r*Tg*CIA1-mAID-HA parasites, shown in schematic (top), and probed in an immunofluorescence assay with anti-Ty1 (green) and anti-*Tg*Tom40 (magenta; mitochondrion) antibodies (bottom). Scale bars is 2 μm. DIC, differential interference contrast. Right, corresponding fluorescence plot depicting the intensity of anti-Ty1 (green) and anti-*Tg*Tom40 (magenta) labeling along the yellow line in merged image. Left, western blot of proteins extracted from r*Tg*CIA1-mAID-HA/c*Tg*CIA1$_{CD5-to-CD1}$-Ty1 parasites, separated by SDS-PAGE and probed with anti-Ty1 antibodies. **(C)** Fluorescence proliferation assays of r*Tg*CIA1-mAID-HA, r*Tg*CIA1-mAID-HA/c*Tg*CIA1$_{WT}$-Ty1, and r*Tg*CIA1-mAID-HA/c*Tg*CIA1$_{CD5-to-CD1}$-Ty1 parasites, cultured in the absence (black) or presence (red) of IAA. Parasite proliferation is expressed as a percentage of the fluorescence measurement in the -IAA

condition on the final day of the assay for each line. Individual data points and error bars represent the mean ± SD of three technical replicates. Error bars not visible are smaller than the symbol. Data are representative of three independent experiments. The numerical data underlying this Figure can be found in S1 Data.

(Fig 7B). We constitutively expressed the resulting protein, which we termed c$Tg$CIA1$_{CD5-to-CD1}$-Ty1, in r$Tg$CIA1-mAID-HA parasites, validated expression by western blotting (Fig 7B), and performed an immunofluorescence assay to determine localization of the protein. Interestingly, we observed that the c$Tg$CIA1$_{CD5-to-CD1}$-Ty1 protein localized exclusively to the mitochondrion, no longer exhibiting the dual localization we observed in the wild type $Tg$CIA1 protein (Fig 7B). We also found that constitutive expression of the c$Tg$CIA1$_{CD5-to-CD1}$-Ty1 protein was unable to rescue the proliferation defect observed upon r$Tg$CIA1-mAID-HA knockdown (Figs 7C and S10D).

Taken together, our data indicate that the mitochondrial targeting of the CTC is mediated by $Tg$CIA1. Specifically, the CD5 loop of $Tg$CIA1 is both necessary and sufficient for mitochondrial targeting, and this targeting is independent of the other mitochondrially-localized CIA pathway proteins $Tg$NBP35, $Tg$Tah18, and $Tg$Nar1. Our data also indicate that, while the position of the CD5 loop in the protein is not critical for mitochondrial targeting, it is critical for facilitating the dual localization of $Tg$CIA1 to the cytosol and mitochondrion. Finally, we have shown that the CD5 loop of $Tg$CIA1, and its positioning within the protein, is critical for $Tg$CIA1 to carry out its functions in parasites.

## A myzozoan-specific amino acid motif in the CD5 loop mediates the mitochondrial localization of $Tg$CIA1

We next set out to uncover the features of the CD5 loop of $Tg$CIA1 that facilitate mitochondrial targeting. Alignments of the CIA1 protein from a range of eukaryotes identified the presence of a short, conserved motif consisting of three aromatic and one positively charged amino acid in the CD5 loop of the CIA1 protein in *T. gondii* and other myzozoans (Figs 8A and S9). This motif (and the extended CD5 loop generally) was not present in other eukaryotic clades, including ciliates such as *P. tetraurelia*, which are the nearest relatives of the myzozoans [14].

We hypothesized that the conserved motif of the CD5 loop could facilitate the mitochondrial targeting of $Tg$CIA1. To test this, we used CRISPR/Cas9-based genome editing to individually substitute each residue in the motif for alanine in the native $Tg$CIA1 locus of the $Tg$CIA1-HA/r$Tg$NBP35-cMyc parasite line. We were successful in generating mutants in the W526, Y527, and R533 residues (but not the F532 residue), terming the resulting proteins $Tg$CIA1$_{W526A}$-HA, $Tg$CIA1$_{Y527A}$-HA, and $Tg$CIA1$_{R533A}$-HA (S12A Fig). SDS-PAGE western blot analyses confirmed that the mutated proteins were expressed at similar abundances to a $Tg$CIA1$_{WT}$-HA control (Figs 8B and S12B). We next performed BN-PAGE western blotting and found that all mutated proteins were present in the >720 kDa CTC as well as in the smaller $Tg$CIA1-containing complexes we had observed previously (Fig 8C).

To test whether the motif is important for mitochondrial targeting, we performed immunofluorescence assays. Notably, all mutations in the CD5 loop motif resulted in the $Tg$CIA1 protein localizing predominantly to the cytosol (Fig 8D), although quantifications revealed that the $Tg$CIA1$_{R533A}$-HA protein exhibited significantly greater mitochondrial co-localization than the $Tg$CIA1$_{W526A}$-HA and $Tg$CIA1$_{Y527A}$-HA proteins (Fig 8E).

Next, we investigated whether the W526, Y527, or R533 residues of the conserved CD5 loop motif contribute to the role of $Tg$CIA1 in parasite proliferation. We performed plaque assays comparing the proliferation of parasites expressing $Tg$CIA1$_{W526A}$-HA, $Tg$CIA1$_{Y527A}$-HA, and $Tg$CIA1$_{R533A}$-HA to parasites expressing $Tg$CIA1$_{WT}$-HA. We found that $Tg$CIA1$_{W526A}$-HA and $Tg$CIA1$_{Y527A}$-HA expressing parasites exhibited severe proliferation defects, indicating that these residues are critical for $Tg$CIA1 function (Fig 8F). By contrast, the proliferation of parasites expressing $Tg$CIA1$_{R533A}$-HA was indistinguishable from those of parasites expressing $Tg$CIA1$_{WT}$-HA, suggesting the R533 residue is largely dispensable for $Tg$CIA1 function (Fig 8F).

We were unable to modify the $Tg$CIA1 locus to express a F532A mutation using the genome editing approach. As an alternative, we constitutively expressed $Tg$CIA1$_{F532A}$-Ty1 from the α-tubulin promoter in r$Tg$CIA1-mAID-HA parasites,

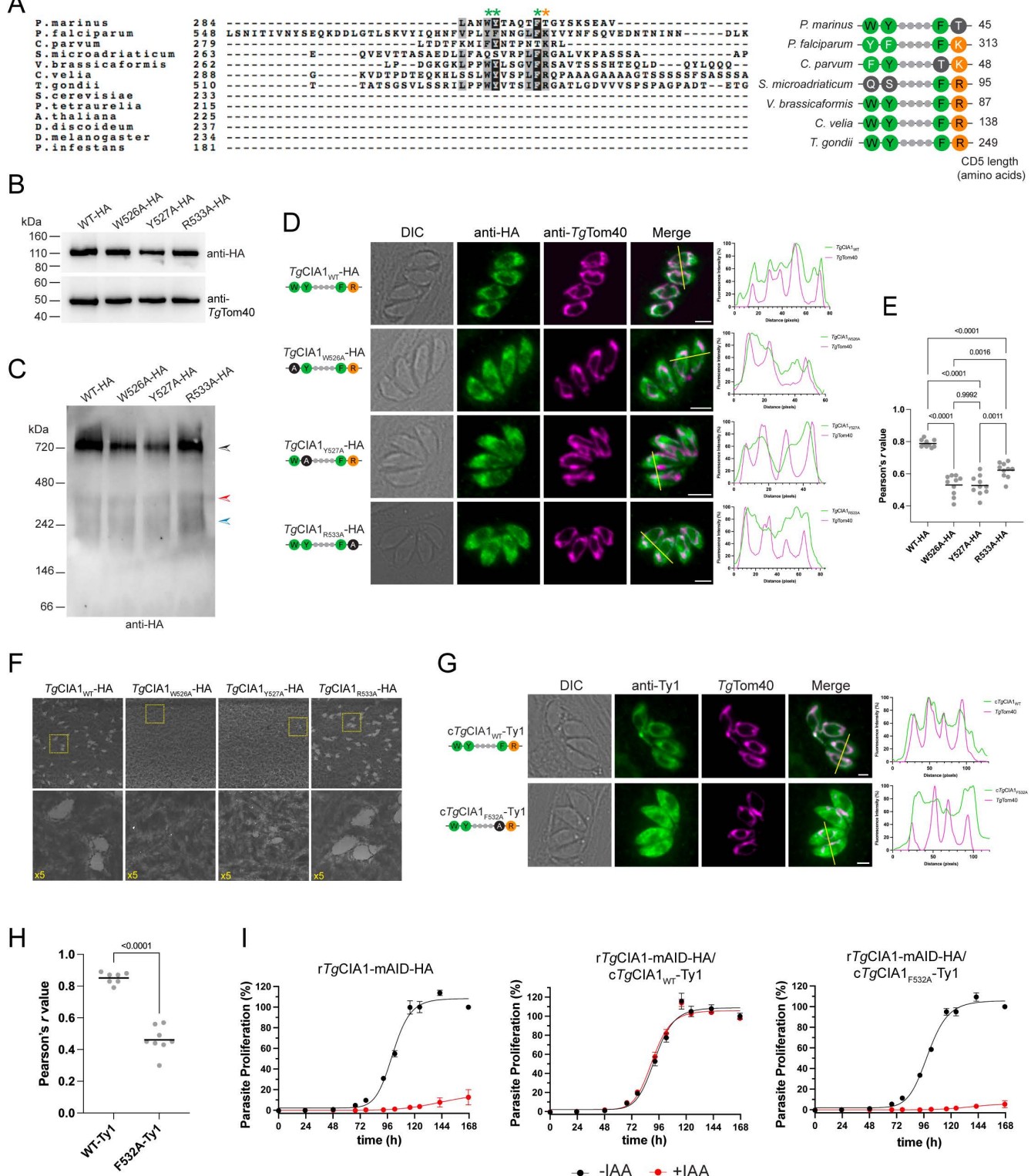

**Fig 8. An aromatic amino acid motif in the CD5 loop of *Tg*CIA1 facilitates mitochondrial targeting. (A)** Left, a region of a multiple sequence alignment of CIA1 homologs highlighting a motif of amino acid residues within the CD5 loop of the protein that is conserved in myzozoans. Asterisks denote key residues of the motif, with green denoting aromatic residues and orange denoting a positively charged residue. Right, a schematic of the motif

observed in the dinozoans *Perkinsus marinus* and *Symbiodinium microadriaticum,* the apicomplexans *Plasmodium falciparum, Cryptosporidium parvum*, and *Toxoplasma gondii,* and the chrompodellids *Vitrella brassicaformis* and *Chromera velia*. Green = aromatic, orange = positively charged, gray = not conserved. Amino acid length of the CD5 loop in each species is listed on the right. **(B, C)** Western blots of proteins extracted from $TgCIA1_{WT}$-HA (WT-HA), $TgCIA1_{W526A}$-HA (W526A-HA), $TgCIA1_{Y527A}$-HA (Y527A-HA), and $TgCIA1_{R533A}$-HA (R533A-HA) expressing parasites, separated by **(B)** SDS-PAGE or **(C)** BN-PAGE, and probed with anti-HA or anti-$Tg$Tom40 antibodies. The black arrowhead indicates the >720 kDa CIA Targeting Complex; red and blue arrowheads indicate the lower mass complexes containing the $Tg$CIA1 protein. **(D)** Immunofluorescence assays of $TgCIA1_{WT}$-HA (WT-HA), $TgCIA1_{W526A}$-HA (W526A-HA), $TgCIA1_{Y527A}$-HA (Y527A-HA), and $TgCIA1_{R533A}$-HA (R533A-HA) parasites. The proteins of interest (green) and the mitochondrion (magenta) were labeled with anti-HA and anti-$Tg$Tom40 antibodies, respectively. Schematics depicting the modified amino acid sequence in the CD5 motif of the proteins from each panel are included next to images (left). Scale bars are 2 µm. DIC, differential interference contrast. Right, corresponding fluorescence profile depicting intensity of anti-HA (green) and anti-$Tg$Tom40 (magenta) labeling along the yellow lines of the merged images. **(E)** The correlation between HA-tagged proteins of interest and $Tg$Tom40 was quantified using the Pearson correlation coefficient ($r$) and the data were analyzed using a one-way ANOVA followed by Tukey's multiple comparisons test. **(F)** Plaque assays of $TgCIA1_{WT}$-HA, $TgCIA1_{W526A}$-HA, $TgCIA1_{Y527A}$-HA, and $TgCIA1_{R533A}$-HA parasites. Parasites were cultured in the absence (top) or presence (bottom) of IAA for 8 days and are representative of three independent experiments. **(G)** Immunofluorescence assays of r$Tg$CIA1-mAID-HA/c$TgCIA1_{WT}$-Ty1 and r$Tg$CIA1-mAID-HA/c$TgCIA1_{F532A}$-Ty1 parasites. The c$TgCIA1_{WT}$-Ty1 protein (green) and the mitochondrion (magenta) were labeled with anti-Ty1 and anti-$Tg$Tom40 antibodies, respectively. Schematics depicting the amino acid sequence in the CD5 motif of the proteins from each panel are included next to images (left). Scale bars are 2 µm. DIC, differential interference contrast. Right, corresponding fluorescence profile depicting intensity of anti-Ty1 (green) and anti-$Tg$Tom40 (magenta) labeling along the yellow lines of the merged images. **(H)** The correlation between Ty1-tagged proteins and $Tg$Tom40 was quantified using the Pearson correlation coefficient ($r$). The data were analyzed using a one-way ANOVA (alongside the $r$ values depicted in S12F Fig) followed by Tukey's multiple comparisons test, with the $p$ value shown. **(I)** Fluorescence proliferation assays of r$Tg$CIA1-mAID-HA, r$Tg$CIA1-mAID-HA/c$TgCIA1_{WT}$-Ty1, and r$Tg$CIA1-mAID-HA/c$TgCIA1_{F532A}$-Ty1 parasites, grown in the absence (black) or presence (red) of IAA. Parasite proliferation is expressed as a percentage of the fluorescence measurement in the -IAA condition on the final day of the assay for each line. Individual data points and error bars represent the mean ± SD of three technical replicates. Error bars not visible are smaller than the symbol. Data are representative of three independent experiments. The numerical data underlying this Figure can be found in S1 Data.

generating a line we termed r$Tg$CIA1-mAID-HA/c$TgCIA1_{F532A}$-Ty1. We found that the c$TgCIA1_{F532A}$-Ty1 protein was expressed at a similar abundance, and in protein complexes of similar masses, to the c$TgCIA1_{WT}$-Ty1 protein (S12C and S12D Fig). Notably, we found that the c$TgCIA1_{F532A}$-Ty1 protein localized predominantly to the cytosol, exhibiting significantly less mitochondrial co-localization than the c$TgCIA1_{WT}$-Ty1 protein (Fig 8G and 8H). We also constitutively-expressed $TgCIA1_{W526A}$-Ty1, $TgCIA1_{Y527A}$-Ty1 and c$TgCIA1_{R533A}$-Ty1 isoforms in r$Tg$CIA1-mAID-HA parasites, and found that the localization of these matched what we observed in the genome-edited point mutants, with all $Tg$CIA1 variants localizing predominantly to the cytosol, although the c$TgCIA1_{R533A}$-Ty1 again exhibited greater mitochondrial co-localization than the other variants (S12E and S12F Fig).

Finally, we undertook fluorescence proliferation assays and plaque assays to test whether the F532 residue of the CD5 loop is important for $Tg$CIA1 function. We compared proliferation of r$Tg$CIA1-mAID-HA/c$TgCIA1_{F532A}$-Ty1 parasites to r$Tg$CIA1-mAID-HA and r$Tg$CIA1-mAID-HA/c$TgCIA1_{WT}$-Ty1 parasite lines cultured in the absence or presence of IAA. We observed that proliferation of r$Tg$CIA1-mAID-HA/c$TgCIA1_{F532A}$ parasites was severely impaired when $Tg$CIA1-mAID-HA was knocked down (Figs 8I and S10E), indicating that the F532 residue is essential for $Tg$CIA1 protein function. We also tested the proliferation of r$Tg$CIA1-mAID-HA/$TgCIA1_{W526A}$-Ty1, r$Tg$CIA1-mAID-HA/$TgCIA1_{Y527A}$-Ty1, and r$Tg$CIA1-mAID-HA/c$TgCIA1_{R533A}$-Ty1 parasites in the absence or presence of IAA. This revealed that r$Tg$CIA1-mAID-HA/c$TgCIA1_{Y527A}$-Ty1 and r$Tg$CIA1-mAID-HA/c$TgCIA1_{R533A}$-Ty1 parasites proliferated normally upon knockdown of the $Tg$CIA1-mAID-HA protein, whereas proliferation of r$Tg$CIA1-mAID-HA/c$TgCIA1_{W526A}$-Ty1 parasites was substantially reduced upon $Tg$CIA1-mAID-HA depletion, although not to the same extent as observed in r$Tg$CIA1-mAID-HA/c$TgCIA1_{F532A}$-Ty1 parasites (S12G and S10E Figs). These findings suggest that constitutive overexpression of the $TgCIA1_{W526A}$ and $TgCIA1_{Y527A}$ mutant isoforms can partially or fully rescue the severe proliferation defects observed in the genome-edited point mutants.

Taken together, these data indicate that the W526, Y527, and F532 residues of the aromatic amino acid motif in the CD5 loop of $Tg$CIA1 are critical for both mitochondrial localization of the $Tg$CIA1 protein and for the functional role of $Tg$CIA1 in parasite proliferation. The R533 residue contributes to the mitochondrial localization of $Tg$CIA1, although not to the same extent as the other residues of this motif that we tested. Surprisingly, despite its role in mitochondrial targeting, the R533 residue of $Tg$CIA1 appears to be dispensable for $Tg$CIA1 function.

Given the conservation of the CD5 loop motif in myzozoans, we asked whether CD5 loops from other myzozoans could complement the function of *T. gondii* CD5 loop. We replaced the CD5 loop of *Tg*CIA1 with the equivalent CD5 loop from the chrompodellid *V. brassicaformis* or the dinozoan *S. microadriaticum*, generating proteins we called c*Tg*CIA1$_{VbCD5}$-Ty1 or c*Tg*CIA1$_{SmCD5}$-Ty1. The *V. brassicaformis* CD5 loop contains the same residues in the conserved aromatic motif as the *Tg*CIA1 protein (Fig 8A), but is considerably shorter than the *T. gondii* CD5 loop (87 amino acids versus 249 amino acids). The *S. microadriaticum* CD5 loop only encodes two amino acids of the motif (the phenylalanine and arginine residues; Fig 8A), and is also considerably shorter than the *T. gondii* CD5 loop (97 amino acids). We expressed the c*Tg*CIA1$_{VbCD5}$-Ty1 and c*Tg*CIA1$_{SmCD5}$-Ty1 proteins in r*Tg*CIA1-mAID-HA parasites, validated expression by SDS-PAGE western blotting (Fig 9A and 9B), and performed immunofluorescence assays to determine protein localization. Both the c*Tg*CIA1$_{VbCD5}$-Ty1 and c*Tg*CIA1$_{SmCD5}$-Ty1 proteins localized predominantly to the cytosol (Fig 9A–9C), although the c*Tg*CIA1$_{SmCD5}$-Ty1 protein also exhibited some observable co-localization with the mitochondrion (Fig 9B, arrowheads).

Next, we investigated whether the CD5 loops of the *Sm*CIA1 and *Vb*CIA1 homologs were functionally equivalent to CD5 loop of *Tg*CIA1. We cultured r*Tg*CIA1-mAID-HA/c*Tg*CIA1$_{VbCD5}$-Ty1 and r*Tg*CIA1-mAID-HA/c*Tg*CIA1$_{SmCD5}$-Ty1 parasites in the absence or presence of IAA and performed fluorescence proliferation and plaque assays. These revealed that both the c*Tg*CIA1$_{SmCD5}$-Ty1- and c*Tg*CIA1$_{VbCD5}$-Ty1-complemented lines proliferated normally when cultured in the presence of IAA (Figs 9D and S10D), indicating that the CD5 loops from the *S. microadriaticum* and *V. brassicaformis* CIA1 proteins can functionally replace the equivalent loop in the *Tg*CIA1 protein of *T. gondii.*

Taken together, our data demonstrate that the CD5 loop of *Tg*CIA1 contains a motif that is conserved throughout the myzozoans (Fig 8A), and which contributes to targeting the CIA1 protein to the mitochondrion. Numerous residues of this motif, including a phenylalanine residue from this motif that is found in all the analyzed myzozoan CIA1 sequences except *Cryptosporidium parvum* (Fig 8A), are critical for *Tg*CIA1 to carry out its biological role.

## The CD5 loop of *Tg*CIA1 is not required for cytosolic protein synthesis

We have shown that the CTC of *T. gondii* exhibits dual localization to the mitochondrion and cytosol, courtesy of a myzozoan-specific CD5 loop in the CIA1 protein of the complex (Fig 10A). It is conceivable that the mitochondrial localization of *Tg*CIA1 is important for the transfer of [4Fe-4S] clusters from mitochondrially-localized Nar1 to the CTC. *Tg*CIA1 localizes to the mitochondrion independently of its association with *Tg*Nar1 (S8H Fig), but it is possible that the CD5 loop of *Tg*CIA1 interacts with mitochondrial outer membrane lipids or an accessory protein of the mitochondrial outer membrane (Fig 10Ai–10Aii). This could place the CTC in position on the outer membrane to interact with *Tg*Nar1 and enable [4Fe-4S] cluster transfer to occur. In these scenarios, the CD5 loop of *Tg*CIA1 functions as a mitochondrial targeting signal to facilitate FeS cluster transfer from *Tg*Nar1. Notably, *T. gondii* expresses the FeS proteins *Tg*ELP3 and *Tg*RlmN on the outer face of outer mitochondrial membrane [31]. An alternative possibility is, therefore, that the CD5 loop instead functions in enabling *Tg*CIA1 and the CTC complex to interact with these client FeS proteins on the outer membrane (Fig 10Aiii). In this scenario, the CD5 loop functions not in enabling [4Fe-4S] cluster transfer from *Tg*Nar1, but instead to enable [4Fe-4S] cluster transfer from the CTC to client mitochondrial outer membrane proteins.

If the mitochondrial localization of *Tg*CIA1 is critical for FeS cluster transfer from *Tg*Nar1, we predicted that impairing the mitochondrial localization of *Tg*CIA1 will impair all downstream processes that require the CTC, such as cytosolic protein translation. We therefore measured protein translation in a parasite strain expressing a *Tg*CIA1 variant that is unable to target to the mitochondrion (c*Tg*CIA1$_{ScCD5}$-Ty1, which lacks the mitochondrial targeting CD5 loop and localizes exclusively to the cytosol; Fig 6C). We compared this to protein translation in a parasite strain expressing a *Tg*CIA1 variant that is targeted exclusively to the mitochondrion (c*Tg*CIA1$_{CD5-to-CD1}$-Ty1; Fig 7B). Remarkably, we still observed robust cytosolic protein translation in parasites expressing only the cytosolically localized c*Tg*CIA1$_{ScCD5}$-Ty1 protein (Fig 10B and 10C). By contrast, we observed a significant depletion of cytosolic protein translation when parasites expressed only the mitochondrially-localized c*Tg*CIA1$_{CD5-to-CD1}$-Ty1 protein (Fig 10B and 10C), similar to the defect we observed when *Tg*CIA1

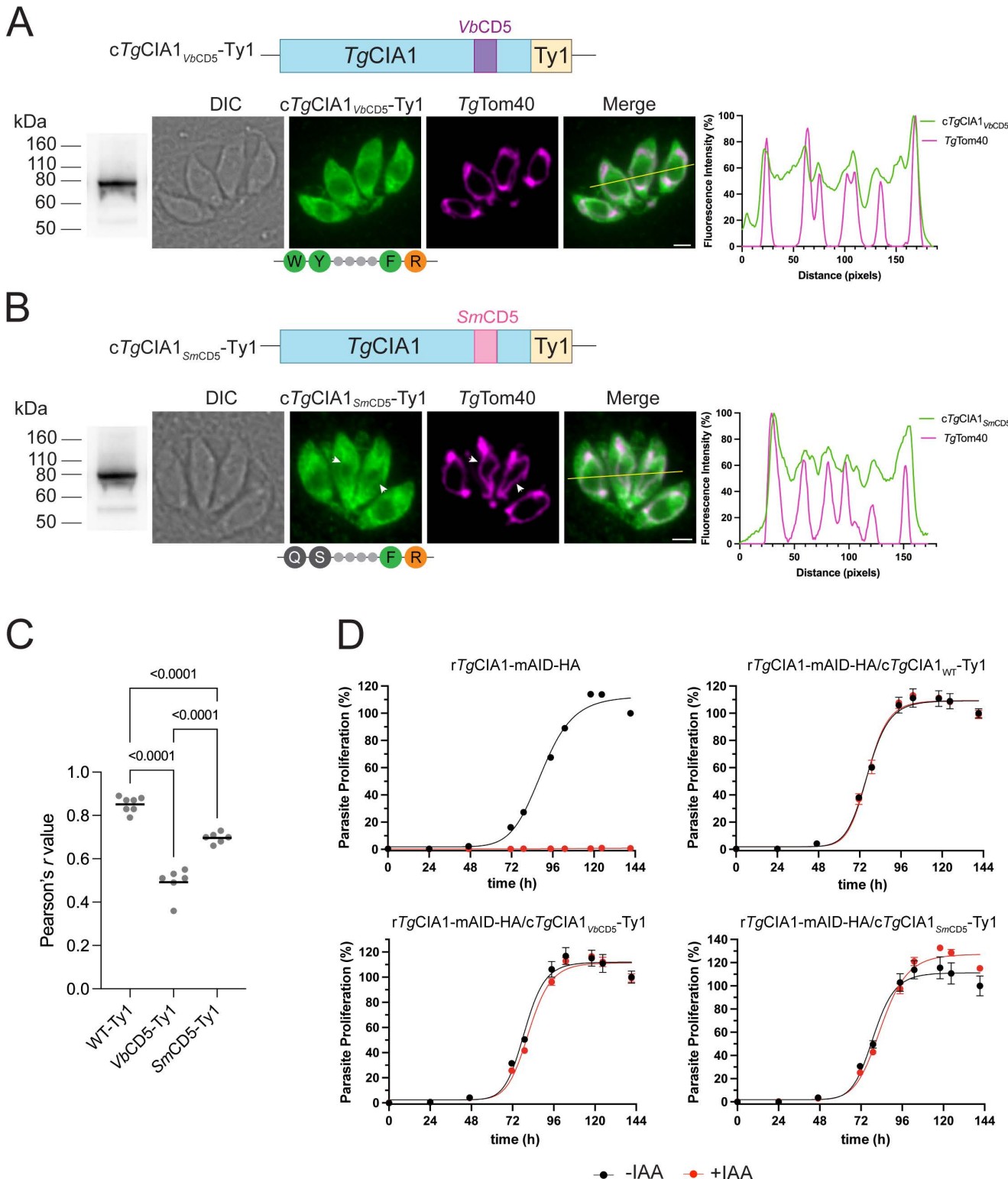

**Fig 9. The CD5 loop of the CIA1 protein is functionally conserved in myzozoans. (A, B)** Immunofluorescence assay of (**A**) rTgCIA1-mAID-HA/cTgCIA1VbCD5-Ty1 and (**B**) rTgCIA1-mAID-HA/cTgCIA1SmCD5-Ty1 parasites, probed with anti-Ty1 (green) and anti-TgTom40 (magenta; mitochondrion) antibodies. Schematics depicting the amino acid sequence in the CD5 motif of the proteins from each panel are included below the Ty1 labeled images. Scale bars are 2 μm. DIC, differential interference contrast. Arrowheads highlight regions where the cTgCIA1SmCD5-Ty1 protein exhibits visible

mitochondrial localization. Right, corresponding fluorescence profile depicting intensity of anti-Ty1 (green) and anti-*Tg*Tom40 (magenta) labeling along the yellow line on merged image. Left, western blots of the proteins of interest, separated by SDS-PAGE and probed with anti-Ty1 antibodies. **(C)** The correlation between the Ty1-tagged proteins of interest and *Tg*Tom40 was quantified using the Pearson correlation coefficient (*r*). The data were analyzed using a one-way ANOVA followed by Tukey's multiple comparisons test with *p* values shown. **(D)** Fluorescence proliferation assays of r*Tg*CIA1-mAID-HA, r*Tg*CIA1-mAID-HA/c*Tg*CIA1$_{WT}$-Ty1, r*Tg*CIA1-mAID-HA/c*Tg*CIA1$_{VbCD5}$-Ty1, and r*Tg*CIA1-mAID-HA/c*Tg*CIA1$_{SmCD5}$-Ty1 parasites, grown in the absence (black) or presence (red) of IAA. Parasite proliferation is expressed as a percentage of the fluorescence measurement in the -IAA condition on the final day of the assay for each line. Individual data points and error bars represent the mean ± SD of three technical replicates. Error bars not visible are smaller than the symbol. Results are representative of three independent experiments. The data for the r*Tg*CIA1mAID-HA and r*Tg*CIA1-mAID-HA/c*Tg*CIA1$_{WT}$-Ty1 lines are identical to those depicted in Fig 7C, the experiments for which were performed simultaneously. The numerical data underlying this Figure can be found in S1 Data.

is depleted (Fig 3C and 3D). These data indicate that CD5 loop-dependent mitochondrial localization of *Tg*CIA1 is not required for the FeS cluster-dependent process of protein translation in the cytosol. These data are, therefore, inconsistent with the hypothesis that the CD5 loop of *Tg*CIA1 is important for FeS cluster transfer from Nar1 to the CTC at the mitochondrial outer membrane (Fig 10Ai–10Aii). Instead, our data indicate that some functions facilitated by *Tg*CIA1 (such as cytosolic protein translation) are independent of the CD5 loop. Our data also suggest that the cytosolic localization of *Tg*CIA1 is required to enable cytosolic protein translation, since we observed a strong translation defect in parasites expressing only the mitochondrially-localized c*Tg*CIA1$_{CD5-to-CD1}$-Ty1 protein, although we cannot rule out this defect is due to the aberrant positioning of the CD5 loop in this protein interfering with CTC function.

## Discussion

In this study, we have examined the localization and importance of the cytosolic FeS cluster assembly (CIA) pathway of *T. gondii* parasites. Our data support a model wherein the CIA pathway of *T. gondii* occurs on the cytosolic face of the mitochondrion (Fig 10A). We propose that, like in other eukaryotes [4,5], a sulfur-containing product (perhaps a [2Fe-2S] cluster) is exported from the mitochondrion. This transport, like in other eukaryotes, involves the mitochondrial inner membrane ABC transporter *Tg*ABCB7L/*Tg*ATM1 [21,22], and could involve outer membrane porins and/or mitoNEET proteins, as proposed recently for animal cells [32]. Homologs of both porins and mitoNEET proteins are encoded in the *T. gondii* genome and localize to the parasite mitochondrion [33,34]. A [4Fe-4S] cluster is then assembled on a NBP35 scaffold, which localizes to the outer face of the outer mitochondrial membrane of *T. gondii* [15].

In other eukaryotes, [4Fe-4S] cluster assembly on the NBP35 scaffold relies on electrons donated from NAPDH via an electron transfer chain consisting of Tah18 and Dre2 [35,36]. We found that *Tg*Tah18 localizes to the mitochondrion, and that its depletion leads to a defect in cytosolic protein translation, which is also observed upon the depletion of most other candidate CIA pathway proteins in *T. gondii* (Figs 3 and S4; [20]). This is consistent with *Tg*Tah18 being a component of the CIA pathway in *T. gondii* (Fig 10A). By contrast, *Tg*Dre2 localizes to the cytosol and is not required for cytosolic protein synthesis, although loss of *Tg*Dre2 does impact the abundance of the cytosolic FeS protein *Tg*ABCE1 (S4 Fig). This suggests that, although *Tg*Dre2 is critical for parasite proliferation, it may not function in the parasite CIA pathway in the same way as the other CIA pathway proteins (Fig 10A). A limitation to this conclusion is that the assays that we have used in this study to measure CIA pathway function, and which have been published in other recent studies [15,20–22], provide only indirect measures for CIA pathway activity (*i.e.,* measure the stability of cytosolic FeS proteins or measure processes that rely on the incorporation of FeS clusters into cytosolic proteins, such as protein translation and lipid droplet biology). A priority for future research in this area will be to develop assays that directly measure the incorporation of FeS clusters into client cytosolic proteins via the CIA pathway in these parasites.

In other eukaryotes, Nar1 is proposed to facilitate the transfer of a [4Fe-4S] cluster from the NBP35 scaffold to the CTC [6,8,26,37]. We found that *Tg*Nar1 is critical for parasite proliferation and cytosolic protein translation, consistent with *Tg*Nar1 functioning in the CIA pathway. This aligns with data from a recent study by Renaud and colleagues, who showed

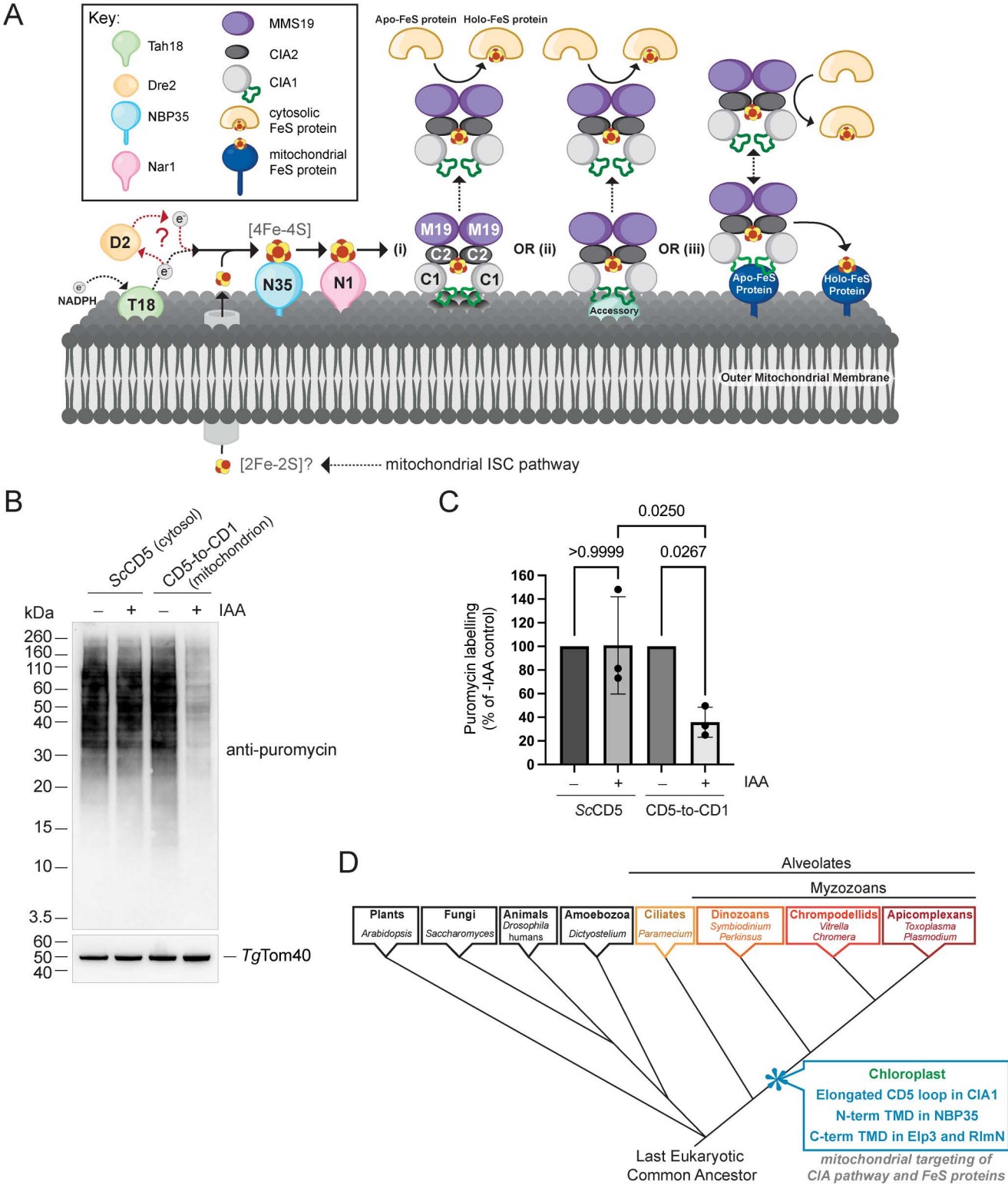

**Fig 10. A model for the spatial organization and evolution of the CIA pathway in _Toxoplasma gondii_ and related eukaryotes. (A)** Models for the spatial organization of the CIA pathway in _T. gondii_ and related organisms. A sulfur-containing product of the mitochondrial ISC pathway (possibly a [2Fe-2S] cluster) is exported from the mitochondrion. [4Fe-4S] clusters assemble on the NBP35 scaffold (N35) at the mitochondrial outer membrane, with electrons for this process donated from NADPH via mitochondrial Tah18 (T18) and possibly Dre2 (D2). [4Fe-4S] clusters are transferred from the NBP35

scaffold to Nar1 (N1), and further to the CIA targeting complex (CTC) comprised of CIA1 (C1), CIA2 (C2) and MMS19 (M19). CIA1 contains an extended CD5 loop (green) that mediates localization of the entire CTC to the mitochondrion. The CD5 loop may bind to (**i**) outer mitochondrial membrane lipids or (**ii**) outer mitochondrial membrane accessory proteins, and place the CTC in position to receive FeS clusters from Nar1. Alternatively, (**iii**) the CD5 loop may mediate interactions with FeS client proteins that are anchored on the mitochondrial outer membrane, thus facilitating FeS transfer from the CTC to these proteins. Elements of the diagram were created in BioRender. Hodgson, E. (2025) https://BioRender.com/ecxz792. **(B)** Western blots measuring the incorporation of puromycin into proteins from r*Tg*CIA1-mAID-HA/c*Tg*CIA1$_{ScCD5}$-Ty1 (*Sc*CD5) or r*Tg*CIA1-mAID-HA/c*Tg*CIA1$_{CD5-to-CD1}$-Ty1 (CD5-to-CD1) parasites cultured in the absence (−) or presence (+) of IAA for 24 h, separated by SDS-PAGE and probed with anti-puromycin and anti-*Tg*Tom40 antibodies. **(C)** Relative abundance of puromycin incorporation from **(B)** was determined as a percentage of the -IAA control for each parasite line, with abundances normalized using the *Tg*Tom40 loading control. Data points represent the mean±SD of three independent experiments. Data were analyzed using a one-way ANOVA followed by Tukey's multiple comparisons test, with relevant *p* values shown. The numerical data underlying this Figure can be found in S1 Data. **(D)** An illustrative phylogenetic tree adapted from currently accepted models of eukaryotic evolution [63–65]. Mitochondrial targeting domains for key components of the CIA machinery, including the extended CD5 loop of CIA1, the N-terminal (N-term) transmembrane domain (TMD) of NBP35 and the C-terminal (C-term) TMDs of the FeS proteins Elp3 and RlmN, all evolved early in myzozoan evolution, subsequent to their divergence from the ciliate lineage. The endosymbiotic acquisition of a chloroplast also occurred early in myzozoan evolution.

that depletion of *Tg*Nar1 resulted in the formation of lipid droplets, a phenotype proposed to result from an impaired CIA pathway [20]. We did not observe a stable interaction between *Tg*Nar1 and the CTC (S8 Fig), although it is likely that FeS cluster transfer between Nar1 and the CTC is a transient process. *Tg*Nar1 contains a conserved C-terminal tryptophan residue that was shown to facilitate interactions between Nar1 and the CTC in mammalian cells (S8 Fig; [25]). Our data indicate that *Tg*Nar1 localizes exclusively to the mitochondrion, suggesting that the transfer of [4Fe-4S] clusters from the *Tg*NBP35 scaffold to the CTC likely takes places on the mitochondrial outer membrane in *T. gondii* (Fig 10A).

The CTC of *T. gondii* parasites exhibits a curious dual localization to both the mitochondrion and cytosol. We show that the mitochondrial association of the CTC is dependent on the *Tg*CIA1 protein of the complex, and that the CD5 loop of *Tg*CIA1 is both necessary and sufficient for its mitochondrial localization. Notably, we identified a conserved aromatic amino acid motif in the loop that is critical for mitochondrial association. Aromatic amino acid residue-containing motifs have been implicated in both protein–protein and protein–lipid interactions [38,39]. We propose that the CD5 motif facilitates the mitochondrial localization of *Tg*CIA1 by associating with either lipids or proteins on the mitochondrial outer membrane (Fig 10Ai–10Aiii).

We found that deleting the CD5 loop or modifying key residues in the aromatic amino acid motif impairs both mitochondrial targeting and *Tg*CIA1 function. This indicates that the CD5 loop, and the aromatic motif therein, plays a critical role in *Tg*CIA1 (and, by extension, CTC) function. One interpretation of these data is that the mitochondrial localization of *Tg*CIA1 is essential for its function. Curiously, however, mutating the arginine residue at position 533 in the aromatic amino acid motif to alanine impairs mitochondrial targeting, but does not ablate protein function (Fig 8 and S12 Fig). It is possible that the aromatic amino acid motif in the CD5 loop plays two independent roles in *Tg*CIA1 biology: a non-essential role in mitochondrial targeting and an essential role in protein function. A limitation of our localization experiments is that they cannot rule out that some mitochondrial association remains in the *Tg*CIA1$_{R533A}$ mutant protein, and indeed quantifications of the extent of overlap between the various point mutants we generated in this study consistently suggest a greater degree of overlap between the *Tg*CIA1$_{R533A}$ protein and the mitochondrion than the other point mutants in the aromatic motif (Figs 8 and S12).

An important remaining question is therefore what role(s) the CD5 loop and aromatic amino acid motif play in *Tg*CIA1 protein function. We considered the possibility that the mitochondrial association of *Tg*CIA1 that is facilitated by the CD5 loop positions the CTC to receive FeS clusters from *Tg*Nar1 at the mitochondrial outer membrane (Fig 10Ai–10Aii). However, we found that parasites expressing a *Tg*CIA1 variant that lacked the CD5 loop (and is impaired in mitochondrial targeting) is still capable of cytosolic protein synthesis (Fig 10B and 10C), a process that relies on a functional CIA pathway [20,21]. This indicates that [4Fe-4S] cluster transfer to cytosolic proteins can still occur in the absence of the CD5 loop of CIA1, suggesting that the CD5 loop of *Tg*CIA1 (and, by extension, the mitochondrial localization of the CTC) is not

required for the maturation of all cytosolic FeS proteins. However, as noted above, our localization studies lack the resolution to definitively conclude that no mitochondrial localization occurs in the CD5 loop mutant.

An alternative possibility is that the CD5 loop of *Tg*CIA1 has a critical role in the transfer of FeS clusters from the CTC to client FeS proteins that reside on the mitochondrial outer membrane (Fig 10Aiii). In this scenario, the CD5 loop functions not as a mitochondrial targeting signal, but in facilitating interactions between the CTC and mitochondrial client proteins. *T. gondii* contains at least two FeS proteins, *Tg*Elp3 and *Tg*RlmN, that localize to the mitochondrial outer membrane courtesy of C-terminal TMDs [31,40]. Both *Tg*Elp3 and *Tg*RlmN are predicted to be important for *T. gondii* proliferation [40–42], so their interactions with the CTC are likely important for parasite survival. In other eukaryotes, [4Fe-4S] cluster transfer from the CTC to Elp3 is proposed to be facilitated by the adaptor protein Elp4 [25,43]. The *T. gondii* genome lacks an Elp4 homolog, and it is possible that the CD5 loop of *Tg*CIA1 instead serves this adaptor role. A recent study identified *Tg*HCF101 as a cytosolic adaptor protein that facilitates FeS cluster transfer from the CTC to *Tg*ABCE1 [20], and our study indicates that the CD5 loop of *Tg*CIA1 is not required for cytosolic protein translation (which is dependent on *Tg*ABCE1), suggesting that different processes may mediate FeS cluster transfer from the CTC to cytosolic and mitochondrial client proteins. Future studies that examine the role of the CD5 loop in *Tg*CIA1 function, and whether it functions in interactions with mitochondrial outer membrane proteins such as *Tg*Elp3 and *Tg*RlmN, will be of particular interest.

The extended CD5 loop of *Tg*CIA is functionally conserved throughout the myzozoan lineage, but absent from ciliates, the closest relatives of the myzozoans (Figs 8, 9, and S9). Intriguingly, the mitochondrial targeting N-terminal TMD of *Tg*NBP35, as well as the C-terminal mitochondrial targeting domains of *Tg*Elp3 and *Tg*RlmN, also appear to have been acquired early in myzozoan evolution (Fig 10D; [15,16,40,41]). Taken together, these observations indicate that the evolution of the myzozoan lineage coincided with migration of the CIA pathway to the mitochondrial outer membrane, and the targeting of some [4Fe-4S] proteins to the mitochondrial outer membrane (Fig 10D). The selective forces that shaped this reorganization of the cytosolic FeS cluster assembly pathway remain unclear. It is conceivable that the relocation of FeS proteins like Elp3 and RlmN to the outer mitochondrial membrane necessitated the relocation of the CIA pathway to the mitochondrion. Interestingly, myzozoan evolution also coincided with the endosymbiotic acquisition of a chloroplast, and the ancestral myzozoan likely relied on photosynthesis for its survival [14]. Iron is an important component of the light harvesting electron transport chains of photosynthesis, and plastids require iron for a range of other important enzymatic processes [44]. The marine environments in which myzozoans evolved is poor in available iron, with iron a major limiting nutrient in the primary productivity of oceans [45,46]. This iron limitation, coupled with the acquisition of an iron-hungry plastid organelle, may have generated a selective pressure to optimize iron utilization in early myzozoans. Locating the CIA pathway to the outer surface of the mitochondrion places it in close proximity to ISC pathway in the mitochondrion, which provides the building blocks for the CIA pathway [4,47]. The mass migration of the CIA pathway to the mitochondrial surface may therefore have evolved to increase the efficiency of iron usage in myzozoans, although the extent to which the localization of the CIA pathway to the mitochondrion increases the efficiency of iron usage is unclear.

Taken together, our study provides a comprehensive analysis of CIA pathway of the apicomplexan parasite *T. gondii*, showing that this pathway is critical for parasite survival. Our data indicate that, in contrast to the CIA pathway of animals, plants and fungi, the mitochondrion of *T. gondii* acts as a hub for the CIA pathway. Our study reveals the evolution of a conserved loop in the CIA1 protein of *T. gondii* and related eukaryotes that facilitates the mitochondrial assoication of the CTC, and which is critical for CIA1 function. The novel features of the CIA pathway that we and others have identified indicate that a ubiquitious and essential metabolic pathway can nevertheless vary considerably between different eukaryotic lineages [15,20].

## Materials and methods

### Parasite culturing

Tachyzoite-stage *T. gondii* parasites were cultured in human foreskin fibroblasts (HFFs) using Dulbecco's Modified Eagle Medium (DMEM) supplemented with 1% (v/v) fetal bovine serum, 2 g/L NaHCO₃, 0.2 mM L-glutamine, 50 units/mL

penicillin, 50 μg/mL streptomycin, 10 μg/mL gentamicin, and 0.25 μg/mL amphotericin b. Cultures were kept in humidified incubators at 37°C and 5% $CO_2$. Where appropriate, 0.1 mM IAA (Merck catalog number I2886), 0.5 μg/mL ATc (Merck catalog number 37919), or an equivalent volume of 100% ethanol (vehicle control) was added.

**Generation of genetically modified parasites**

We used CRISPR/Cas9 genome editing to introduce 3× HA, 3′ mini-auxin-inducible degron-3× HA (mAID-HA), 3′ mAID-3× cMyc (mAID-cMyc), 5′ HA-mAID, 3′ Ty1, or 3′ spaghetti monster fluorescent protein (smFP)-cMyc epitope tags into the 3′ or 5′ regions of the open reading frames of *Tg*Tah18 (www.ToxoDB.org identifier TGGT1_249320; [17]), *Tg*Dre2 (TGGT1_216900), *Tg*Nar1 (TGGT1_242580) *Tg*CIA1 (TGGT1_313280), *Tg*CIA2 (TGGT1_306590), *Tg*MMS19 (TGGT1_222230), or *Tg*ABCE1 (TGGT1_216790). To enable, these modifications, we introduced DNA encoding single guide (sg)RNAs targeting the 3′ or 5′ regions of these genes into the vector pSAG1::Cas9-U6-UPRT (Addgene plasmid 54467, [48]) using Q5-site directed mutagenesis (New England Biolabs). We performed the Q5 reaction using the gene-specific "3′ or 5′ CRISPR fwd" primer and a "universal CRISPR rvs" primer (S1 Table). We amplified a donor DNA sequence encoding the epitope tag plus 50 bp flanks of the target gene using gene specific "tag fwd" and "tag rvs" primers and HA, C-terminal mAID-HA, N-terminal HA-mAID, mAID-cMyc, cMyc, Ty1, or smFP-cMyc gBlocks (Integrated DNA Technologies, IDT) as templates (S1 Table).

We transfected the sgRNA-expressing plasmids and HA-containing donor DNAs targeting the 3′ region of the *Tg*Tah18, *Tg*Dre2, *Tg*CIA1, *Tg*CIA2, or *Tg*MMS19 into an ATc regulatable *Tg*NBP35-cMyc strain described previously [15]. This generated *Tg*Tah18-HA, *Tg*Dre2-HA, *Tg*CIA1-HA, *Tg*CIA2-HA, and *Tg*MMS19-HA expressing parasites. We transfected the sgRNA-expressing plasmids targeting *Tg*Tah18, *Tg*Dre2, *Tg*Nar1, *Tg*CIA1, *Tg*CIA2, or *Tg*MMS19 and C-terminal mAID-HA, N-terminal HA-mAID, or mAID-cMyc-containing donor DNAs into RHΔ*ku80*-Tir1-FLAG parasites expressing tdTomato red fluorescent protein (described previously, [18]). This generated the IAA-regulatable r*Tg*Tah18-mAID-HA, r*Tg*Dre2-mAID-HA, rHA-mAID-*Tg*Nar1, r*Tg*CIA1-mAID-HA, r*Tg*CIA2-mAID-HA, r*Tg*MMS19-mAID-HA, and r*Tg*CIA1-mAID-cMyc parasite lines. We transfected the sgRNA-expressing plasmid targeting *Tg*ABCE1 and the corresponding cMyc donor DNA into the r*Tg*CIA1-mAID-HA and r*Tg*Dre2-mAID-HA parasite lines to generated the r*Tg*CIA1-mAID-HA/*Tg*ABCE1-cMyc and r*Tg*Dre2-mAID-HA/*Tg*ABCE1-cMyc lines, respectively. We transfected the sgRNA-expressing plasmids targeting *Tg*CIA2 and *Tg*MMS19 and the corresponding HA donor DNA into the r*Tg*CIA1-mAID-cMyc parasite line to generate the r*Tg*CIA1-mAID-cMyc/*Tg*CIA1-HA and r*Tg*CIA1-mAID-cMyc/*Tg*MMS19-HA lines. We transfected the sgRNA-expressing plasmid targeting *Tg*CIA1 and the corresponding smFP-cMyc donor DNA into the r*Tg*CIA2-mAID-HA, r*Tg*MMS19-mAID-HA, r*Tg*Tah18-mAID-HA, and rHA-mAID-*Tg*Nar1 parasite lines to generate the r*Tg*CIA2-mAID-HA/*Tg*CIA1-smFP-cMyc, r*Tg*MMS19-mAID-HA/*Tg*CIA1-smFP-cMyc, r*Tg*Tah18-mAID-HA/*Tg*CIA1-smFP-cMyc, and rHA-mAID-*Tg*Nar1/*Tg*CIA1-smFP-cMyc lines, respectively.

Two days after each of the transfections described above, GFP-positive parasites (which express Cas9-GFP encoded on the modified pSAG1::Cas9-U6 vector) were sorted into wells of a 96-well plate using a FACSMelody cell sorter (BD Biosciences). After approximately one week, we identified wells containing clonal parasites (*i.e.,* wells with single plaques), extracted genomic DNA from parasites in these wells, and performed PCR screening analyses to identify genetically modified parasites using gene-specific "3′ or 5′ scrn fwd" and "3′ or 5′ scrn rvs" primers (S1 Table).

To complement the r*Tg*CIA2-mAID-HA line with a constitutively-expressed copy of wild type *Tg*CIA2, we synthesized the *Tg*CIA2$_{WT}$ open reading frame and an in-frame 3′ Ty1 epitope tag as a gBlock (S1 Table). We PCR amplified the *Tg*CIA2$_{WT}$-Ty1 open reading frame with the primers CIA2 comp fwd and Ty1 rvs, digested the resulting PCR product with *Bgl*II and *Xma*I and ligated this into the equivalent restriction enzyme cut sites of pUDTG [49]. The resulting vector, which we termed pUDTTy(*Tg*CIA2$_{WT}$), expresses *Tg*CIA2$_{WT}$-Ty1 from the constitutive α-tubulin promoter, in a vector that contains a pyrimethamine-resistant *Tg*DHFR selectable marker and a flanking sequence of the non-essential *Tg*UPRT gene for genomic integration. To complement the r*Tg*CIA2-mAID-HA line with a constitutively-expressed copy of *Tg*CIA2 in which

the putative FeS cluster-binding residue Cys-524 was mutated to Ala, we synthesized the $TgCIA2_{C524A}$ open reading frame and an in-frame 3′ Ty1 tag as a gBlock (S1 Table). We PCR amplified the $TgCIA2_{C524A}$-Ty1 open reading frame with the primers CIA2 comp fwd and Ty1 rvs, digested the resulting PCR product with *Bgl*II and *Avr*II (excising the $TgCIA2_{C524A}$ open reading frame) and ligated this into the equivalent sites of pUDTTy($TgCIA2_{WT}$) generating a plasmid we termed pUDTTy($TgCIA2_{C524A}$). We linearized the pUDTTy($TgCIA2_{WT}$) and pUDTTy($TgCIA2_{C524A}$) vectors with *Mfe*I, transfected into r$TgCIA2$-mAID-HA parasites, selected on 1 μM pyrimethamine and cloned parasites by limiting dilution. This generated the r$TgCIA2$-mAID-HA/c$TgCIA2_{WT}$-Ty1 and r$TgCIA2$-mAID-HA/c$TgCIA2_{C524A}$-Ty1 parasite lines.

To complement the r$TgCIA1$-mAID-HA line with constitutively-expressed copies of wild type $TgCIA1$ ($TgCIA1_{WT}$), $TgCIA1$ with the CD1 loop replaced with the CD1 loop from yeast ($TgCIA1_{ScCD1}$), $TgCIA1$ with the CD5 loop replaced with the CD5 loop from yeast ($TgCIA1_{ScCD5}$), $TgCIA1$ with the CD1 loop replaced with the *T. gondii* CD5 loop and the native *T. gondii* CD5 loop replaced with the CD5 loop from yeast ($TgCIA1_{CD5-to-CD1}$), $TgCIA1$ with the CD5 loop replaced with the CD5 loop from *V. brassicaformis* ($TgCIA1_{VbCD5}$), or $TgCIA1$ with the CD5 loop replaced with the CD5 loop from *S. minutum* ($TgCIA1_{SmCD5}$), we synthesized the open reading frames of the desired proteins as gBlocks (S1 Table). We PCR amplified the $TgCIA1$ variants with the primers CIA1 comp fwd and Ty1 rvs, digested the resulting PCR product with *Bgl*II and *Avr*II, and ligated this into the equivalent restriction enzyme sites of the pUDTTy($TgCIA2_{WT}$) vector. This generated vectors we termed pUDTTy($TgCIA1_{variant}$). We linearized the resulting vectors with *Mfe*I, transfected into r$TgCIA1$-mAID-HA parasites, selected on 1 μM pyrimethamine, and cloned parasites by limiting dilution. This generated the r$TgCIA1$-mAID-HA/c$TgCIA1_{CD\ variant}$-Ty1 cell lines.

To complement the r$TgCIA1$-mAID-HA line with constitutively-expressed copies of $TgCIA1$ with the CD3 loop replaced with the CD3 loop from yeast ($TgCIA1_{ScCD3}$), or where residues in the CD5 aromatic amino acid motif were modified to alanines ($TgCIA1_{W526A}$, $TgCIA1_{Y527A}$, $TgCIA1_{F532A}$, or $TgCIA1_{R533A}$), we undertook a site-directed mutagenesis approach. We initially digested the pUDTTy($TgCIA1_{WT}$) vector with *Spe*I and *Nsi*I, and religated the vector using the compatible sticky ends generated in the digest. This removed a large portion of the vector (the DHFR selectable marker and some of the UPRT flank), and made the resulting $TgCIA1$-encoding vector smaller and, consequently, easier to use for site-directed mutagenesis. We performed Q5 site-directed mutagenesis with the primers $CIA1_{ScCD3}$, $CIA1_{W526A}$, $CIA1_{Y527A}$, $CIA1_{F532A}$, or $CIA1_{R533A}$ fwd and rvs as per the manufacturer's instructions (New England Biolabs). After verifying successful mutagenesis by Sanger sequencing, we excised the regions of the resulting vectors encoding the $TgCIA1$ variant with *Bgl*II and *Avr*II, and ligated these into the equivalent sites of the pUDTTy($TgCIA2_{WT}$) vector. This generated vectors we termed pUDTTy($TgCIA1_{variant}$). We linearized the resulting vectors with *Mfe*I, transfected into r$TgCIA1$-mAID-HA parasites, selected on 1 μM pyrimethamine, and cloned parasites by limiting dilution. This generated the r$TgCIA1$-mAID-HA/c$TgCIA1_{ScCD3}$-Ty1 and r$TgCIA1$-mAID-HA/c$TgCIA1_{aromatic\ motif\ mutant}$-Ty1 cell lines.

To constitutively express GFP containing the CD5 loop of $TgCIA1$ ($GFP_{TgCD5}$), we synthesized a GFP-$TgCD5$ gBlock and PCR amplified the GFP-$TgCD5$ open reading frame with the primers GFP fwd and rvs (S1 Table). We digested the resulting PCR product with *Bgl*II and *Avr*II, and ligated this into the equivalent restriction enzyme sites of the pUDTTy($TgCIA2_{WT}$) vector. This generated vectors we termed pUDTTy($GFP_{TgCD5}$). We linearized the resulting vector with *Mfe*I, transfected into RHΔ*ku80*-Tir1-FLAG/tdTomato parasites, and selected on 1 μM pyrimethamine.

To constitutively-express wild type *D. melanogaster* CIA1 ($DmCIA1_{WT}$) or $DmCIA1$ with the CD5 loop replaced with the CD5 loop from *T. gondii* ($DmCIA1_{TgCD5}$), we synthesized both variants as gBlocks, then PCR amplified them using the primers $DmCIA1$ fwd and Ty1 rvs. We digested the resulting PCR products with *Bgl*II and *Avr*II, and ligated them into the equivalent restriction enzyme sites of the pUDTTy($TgCIA2_{WT}$) vector. This generated vectors we termed pUDTTy($DmCIA1_{WT/TgCD5}$). We linearized the resulting vectors with *Mfe*I, transfected into r$TgCIA1$-mAID-HA parasites, selected on 1 μM pyrimethamine, and cloned parasites by limiting dilution. This generated the r$TgCIA1$-mAID-HA/c$DmCIA1_{WT/TgCD5}$-Ty1 cell lines.

To engineer point mutations in the aromatic amino acid motif of the CD5 loop of the native $TgCIA1$ locus of *T. gondii*, we adopted a CRISPR/Cas9-based genome editing approach. We introduced DNA encoding sgRNAs targeting the

genome near the region encoding the *Tg*CIA1$_{W526}$ and *Tg*CIA1$_{Y527}$ residues or near the region encoding the *Tg*CIA1$_{F532}$ and *Tg*CIA1$_{R533}$ residues into the pSAG1::Cas9-U6-UPRT vector using Q5-site directed mutagenesis (New England Biolabs). We performed the Q5 reaction using the gene specific "CIA1-W526/Y527 CRISPR fwd" or "CIA1-F532/R533 CRISPR fwd" primers and a "universal CRISPR rvs" primer (S1 Table). To generate repair templates, we annealed complementary oligo-nucleotides termed W526A fwd and rvs, Y527A fwd and rvs or R533A fwd and rvs (S1 Table) by mixing 2 nmol of the fwd and rvs oligos then heating to 98°C for 3 min, before allowing the samples to cool to room temperature. We co-transfected the annealed W526A or Y527A oligos with the plasmid encoding the CIA1-W526/Y527-targeting sgRNA, and the annealed R533A oligos with the plasmid encoding the CIA1-F532/R533 sgRNA, into r*Tg*NBP35-cMyc/*Tg*CIA1-HA parasites, then sorted and cloned GFP-positive parasites 2 days post-transfection. We extracted genomic DNA from resulting clones and amplified the region encoding the *Tg*CIA1 aromatic amino acid motif with the primers CIA1 motif seq fwd and rvs (S1 Table), and identified clones harboring successful genetic modifications by Sanger sequencing of the resulting PCR products.

Transfections were performed as described previously [50], using 2 mm gap cuvettes and a single 1.5 kV pulse at 25 μF capacitance and 50 Ω resistance using a Bio-Rad Gene Pulser II electroporator.

## Immunofluorescence assays

Confluent HFFs growing on glass coverslips were inoculated with freshly egressed parasites and cultured for ~24 hours at 5% $CO_2$ and 37°C. Infected host cells were then fixed in 3% (v/v) paraformaldehyde in phosphate-buffered saline (PBS; 137 mM NaCl, 2.7 mM KCl, 10 mM $Na_2HPO_4$, 1.8 mM $KH_2PO_4$, pH 7.4) for 15–20 min and permeabilized in 0.25% (v/v) Triton X-100 (TX-100) in PBS for 10 min. Samples were blocked in 2% or 4% (w/v) bovine serum albumin (BSA) in PBS overnight at 4°C. Samples were probed with either rat anti-HA (1:200 dilution; clone 3F10 Sigma-Aldrich, catalog number 11867423001), mouse anti-cMyc (1:100 or 1:200 dilution; clone 9E10 Santa Cruz Biotechnology, catalog number SC-40) or mouse anti-Ty1 (1:200 dilution; clone BB2, [51]) primary antibodies, together with rabbit anti-*Tg*Tom40 (1: 2,000 dilution; [52]) primary antibody. Samples were subsequently probed with donkey anti-rat AlexaFluor 488 (1:500 dilution; Thermo Fisher Scientific, catalog number A21208), goat anti-mouse AlexaFluor 488 (1:250 or 1:500 dilution; Thermo Fisher Scientific, catalog number A11029), or goat anti-mouse AlexaFluor 546 (1:250 dilution; Thermo Fisher Scientific, catalog number A11030) secondary antibodies together with goat anti-rabbit AlexaFluor 546 (1:500 dilution; Thermo Fisher Scientific, catalog number A11035) or goat anti-rabbit AlexaFluor 647 (1:500 dilution; Thermo Fisher Scientific, cat-alog number A21245) secondary antibodies. Images were acquired on a DeltaVision Elite deconvolution microscope (GE Healthcare) using a Photometrics CoolSNAP HQ$^2$ camera. Brightness and contrast were adjusted linearly using SoftWoRx suit 2.0 software. Images were artificially colored and the fluorescence intensity profiles, measuring pixel intensity along a line of interest, were generated using FIJI software [53]. The fluorescence intensity values were normalized to the max-imum value of either channel using the equation *x normalized = (x − x minimum)/(x maximum − x minimum)* and plotted using GraphPad Prism 9 software. To analyze the degree of signal overlap between two channels within the boundary of a parasite vacuole Pearson's correlation coefficients were determined. Pearson's *r* values were generated using the plugin Coloc2 with Costes regression using FIJI software [53] and plotted using GraphPad Prism 10 software. To test the extent of non-specific antibody labeling, we performed immunofluorescence assays on TATiΔ*ku80* parasites lacking epitope tags. These data revealed minimal levels of non-specific labeling with the epitope tag antibodies at the antibody concentrations and imaging conditions that we used in the study (S13 Fig).

## SDS-PAGE, BN-PAGE, and western blotting

For SDS-PAGE, parasite proteins were solubilized in LDS sample buffer (Thermo Fisher Scientific) containing 2.5% v/v β-mercaptoethanol to a concentration of $2.5 \times 10^5$ parasites/μL. Proteins from $2.5 \times 10^6$ parasite equivalents were

separated on a pre-cast NuPage 12% Bis-Tris polyacrylamide gel (Thermo Fisher Scientific) and transferred onto a 0.45 μm pore-sized nitrocellulose membrane.

For BN-PAGE, parasite proteins were solubilized in BN-PAGE lysis buffer (Thermo Fisher Scientific NativePAGE sample buffer containing 1% v/v TX-100, 2 mM EDTA, and 1× cOmplete protease inhibitor cocktail, Merck, catalog number 11873580001) to a concentration of $2.5 \times 10^5$ parasites/μl. Proteins from $2.5 \times 10^6$ parasite equivalents were separated on a pre-cast NativePAGE 4%–16% Bis-Tris polyacrylamide gel (Thermo Fisher Scientific), transferred onto a polyvinylidene difluoride (PVDF) membrane, fixed in 10% (v/v) acetic acid, and de-stained in methanol.

Nitrocellulose and PVDF membranes were blocked overnight in "Blotto," a Tris-buffered saline (TBS; 137 mM NaCl, 2.7 mM KCl, 25 mM Tris-HCl, pH 7.4) solution containing 4% (w/v) skim milk powder. Membranes were subsequently probed with rat anti-HA (1:200–1:400 dilution; clone 3F10 Sigma-Aldrich, catalog number 11867423001), mouse anti-cMyc (1:100 or 1:200 dilution; clone 9E10 Santa Cruz Biotechnology, catalog number SC-40), mouse anti-Ty1 (1:200 or 1:400 dilution; clone BB2 [51]), mouse-anti-puromycin (1:3,000 dilution; clone 12D10, Merck catalog number MABE343), or rabbit anti-*Tg*Tom40 (1:2,000 dilution [52]) primary antibodies diluted in Blotto. Secondary antibodies used were horseradish peroxidase-conjugated goat anti-rat IgG (1:5,000 or 1:10,000 dilution; Abcam, catalog number ab97057), goat anti-mouse IgG (1:2,500, 1:5,000, or 1:10,000 dilution; Abcam, catalog number ab6789), or goat anti-rabbit IgG (1:5,000 or 1:10,000 dilution; Abcam, catalog number ab97051).

Antibody-labeled membranes were incubated in enhanced chemiluminescence solution (0.04% w/v luminol, 0.007% w/v coumaric acid, 0.01% $H_2O_2$, 100 mM Tris pH 9.35) and imaged using a ChemiDoc MP imaging system (Bio-Rad).

In western blotting experiments comparing the expression of different proteins in a particular parasite line (*e.g.,* where band intensities were quantified), the same membrane was probed multiple times with different antibodies, with bound antibodies removing by stripping between probings. For stripping, membranes were treated twice in stripping buffer (200 mM glycine, 3.5 mM SDS, 1% v/v Tween-20, pH 2) for 15 min, twice in PBS for 10 min, twice in TBS containing 0.05% (v/v) Tween-20 for 5 min, and then blocked in Blotto for at least 1 h. Where relevant, band intensities were quantified using ImageJ (1.53k; [54]).

### Puromycin incorporation assays

Puromycin incorporation assays were adapted from a previously described methodology [21]. Parasites were cultured in the absence or presence of IAA for 24 h. Intracellular parasites were mechanically egressed from host cells by passage through a 26-gauge needle, and host cell debris removed by filtering samples through a 5 μm PVDF filter. Parasites were pelleted by centrifugation at 1,500*g* and resuspended in pre-warmed complete culture medium (±IAA) to a final concentration of approximately $1.5 \times 10^7$ parasites/mL. Puromycin (Merck, catalog number P8833) was added to the parasite samples to a final concentration of 10 μg/mL, and parasites were incubated in a humidified 37°C incubator at 5% $CO_2$ for 15 min with loosened lids to allow gas exchange. Puromycin-labeled parasites were pelleted by centrifugation at 12,000*g* for 1 min, washed in room temperature PBS, re-pelleted by centrifugation at 12,000*g* for 1 min, and resuspended in reducing LDS sample buffer (Thermo Fisher Scientific) to a final concentration of $2.5 \times 10^5$ parasites/μl. Proteins were separated by SDS-PAGE and detected by western blotting as described above.

### Co-immunoprecipitations

Immunoprecipitations were performed as described previously [49,52]. Briefly, parasites were solubilized on ice for at least 30 min in a lysis buffer containing 1% (v/v) TX-100, 150 mM NaCl, 2 mM EDTA, 1× cOmplete protease inhibitor cocktail, and 50 mM Tris-HCl (pH 7.4). Insoluble proteins were removed by centrifugation at 21,000*g* and the clarified supernatant was incubated overnight with anti-HA affinity matrix (Sigma-Aldrich, catalog number 11815016001) or Myc-Trap anti-cMyc agarose beads (Chromotek, catalog reference yta) at 4°C on a spinning wheel. Unbound proteins were precipitated with

trichloroacetic acid. Beads containing bound proteins were washed in a wash buffer containing 0.01% or 1% v/v TX-100, 150 mM NaCl, 2 mM EDTA, 50 mM Tris-HCl pH 7.4, and proteins were eluted from the beads using LDS sample buffer (Thermo Fisher Scientific) containing 2.5% v/v β-mercaptoethanol. The unbound and bound fractions were resuspended in reducing LDS sample buffer, with protein samples subsequently separated by SDS-PAGE and detected by western blotting as described above.

**Fluorescence proliferation assays**

Fluorescence proliferation assays were performed as described previously [55]. Optical bottom 96-well plates (Corning or Greiner) containing confluent HFFs were inoculated with 2,000 tdTomato-expressing parasites, and cultured in the absence or presence of IAA in phenol-red-free DMEM. Each condition was performed in triplicate. Well fluorescence was analyzed using a FLUOstar Optima plate reader (BMG Labtech) once daily, or twice daily during the expected exponential growth phase, over a 5–6 day period. Fluorescence values (with average background fluorescence subtracted) were expressed as a percentage of the average final fluorescence measurement in the -IAA condition for each cell line and were plotted using GraphPad Prism 9 or 10, with a sigmoidal curve fitted to the data using the equation:

$$Y = Bottom + \left(X^{Hillslope}\right) * \frac{Top-Bottom}{X^{Hillslope} + EC50^{Hillslope}}.$$

**Plaque assays**

Confluent HFFs in 25 cm$^2$ tissue culture flasks were inoculated with 500 parasites and cultured in the absence or presence of IAA for 6 to 8 days at 37°C and 5% $CO_2$. Infected HFFs were stained with Gram's crystal violet solution (Merck, catalog number 94448) for 1–3 hours and rinsed with PBS. Flasks were air-dried before imaging with a CanoScan 9000F scanner (Canon).

**Multiple sequence alignment and structural analyses**

CIA1 homologs were identified by screening the CIA1 ortholog group (OG6_102166) on OrthoMCL (https://orthomcl.org/orthomcl; [56]). If a CIA homolog sequence was not found in an organism-of-interest on OrthoMCL, additional BLAST searches were conducted on NCBI (https://blast.ncbi.nlm.nih.gov/Blast.cgi; [57]) or UniProt (https://www.uniprot.org; [58]). We performed reciprocal BLAST searches with each candidate hit on ToxoDB (www.toxodb.org; [17]) and excluded sequences where *Tg*CIA1 was not recognized as the top hit. Clustal Omega (https://www.ebi.ac.uk/Tools/msa/clustalo/; [59]) was used to generate a multiple sequence alignment. Sequences that exhibited significant misalignment were excluded. The alignments were plotted using pyBoxshade (https://github.com/mdbaron42/pyBoxshade). Multiple sequence alignments of *Tg*CIA2 and *Tg*Nar1 with homologs from other organisms were also performed using Clustal Omega, and plotted using pyBoxshade, with protein sequences acquired from VEuPathDB (www.veupathdb.org; [60]), ToxoDB and UniProt. The CIA2 alignment was manually adjusted using Jalview (version 2.11.4.1; [61]).

The structure of *S. cerevisiae* CIA1 was generated previously [28] and was obtained from the Protein Data Bank (PDB: 2HES). The predicted structure of *Tg*CIA1 was obtained from AlphaFold2 (UniProt ID A0A125YRQ0; [62]). The structural analyses of CIA1 homologs were viewed and colored using EzMol software (http://www.sbg.bio.ic.ac.uk/ezmol/; [63]).

**Data analysis and availability**

All statistical tests were performed in GraphPad Prism 9 or 10 software and are described in the figure legends. Most fluorescence proliferation assays, plaque assays and western blots depicted in the figures are representative of at least three independent experiments (as indicated in the figure legends). These replicate proliferation, plaque assay and western blotting data, as well as additional representative immunofluorescence images from the microscopy experiments, are

included in S2 Data. Source images for all western blots, PCRs, and plaque assays are included in a S1 Raw Images, and the numerical values (both raw and normalized) for the graphed data are included in the S1 Data.

## Supporting information

**S1 Fig. Epitope tagging of candidate CIA pathway proteins in *Toxoplasma gondii*. (A–E)** 3× hemagglutinin epitope tags (HA) were integrated into the 3′ regions of the open reading frames of the genes encoding **(A)** *Tg*Tah18, **(B)** *Tg*Dre2, **(C)** *Tg*CIA1, **(D)** *Tg*CIA2, or **(E)** *Tg*MMS19 in ATc-regulatable r*Tg*NBP35-cMyc parasites. **(F)** A 3× hemagglutinin-mini-auxin inducible degron (HA-mAID) epitope tag was integrated into the 5′ region of the *Tg*Nar1 open reading frame in RHΔ*ku80*/Tir1-FLAG/tdTomato parasites, generating the rHA-mAID-*Tg*Nar1 parasite line. A schematic depicting the target locus before and after modification, the approximate position of the forward and reverse primers used in the PCR analysis, and the expected sizes of the PCR products in the native and modified genomic loci, are shown at the top of each panel. The PCR screens testing for genetic modifications are shown at the bottom of each panel. PCRs were performed using forward and reverse primers specific to the target site of each gene, and using genomic DNA extracted from clonal parasite lines. Genomic DNA from a wild type (WT) parasite line was used as a control for the expected size of the native locus in each screen.
(TIF)

**S2 Fig. Generating IAA-regulatable strains of candidate CIA pathway proteins in *Toxoplasma gondii*. (A–E)** Mini-auxin inducible degron-3× hemagglutinin (mAID-HA) epitope tags were integrated into the 3′ regions of open reading frames of the genes encoding **(A)** *Tg*Tah18, **(B)** *Tg*Dre2, **(C)** *Tg*CIA1, **(D)** *Tg*CIA2, or **(E)** *Tg*MMS19 in RHΔ*ku80*/Tir1-FLAG/tdTomato parasites generating the r*Tg*Tah18-mAID-HA, r*Tg*Dre2-mAID-HA, r*Tg*CIA1-mAID-HA, r*Tg*CIA2-mAID-HA, and r*Tg*MMS19-mAID-HA parasite lines. A schematic depicting the target locus before and after modification, the approximate position of the forward and reverse primers used in the PCR analysis, and the expected sizes of the PCR products in the native and modified genomic loci, are shown at the top of each panel. The PCR screens testing for genetic modifications are shown at the bottom of each panel. PCRs were performed using forward and reverse primers specific to the target site of each gene, and using genomic DNA extracted from clonal parasite lines. Genomic DNA from a WT parasite line was used as a control for the expected size of the native locus in each screen.
(TIF)

**S3 Fig. Epitope tagging *Tg*ABCE1 in the r*Tg*CIA1-mAID-HA and r*Tg*Dre2-mAID-HA lines. (A)** A schematic depicting the *Tg*ABCE1 genomic locus before and after introduction of a 3× cMyc epitope tag into the 3′ region of the open reading frame of the gene, the approximate position of the forward and reverse primers used in the PCR analyses, and the expected sizes of the PCR products in the native and modified *Tg*ABCE1 loci. **(B, C)** PCR analyses using the forward and reverse primers and template genomic DNA extracted from **(B)** r*Tg*CIA1-mAID-HA/*Tg*ABCE1-cMyc parasite clones and **(C)** a r*Tg*Dre2-mAID-HA/*Tg*ABCE1-cMyc parasite clone. Genomic DNA from a WT parasite line was used as a control for the expected size of the native locus.
(TIF)

**S4 Fig. Depletion of most candidate CIA pathway proteins leads to a decrease in protein translation in *Toxoplasma gondii* parasites. (A)** Western blots measuring the incorporation of puromycin into proteins from r*Tg*Tah18-mAID-HA, r*Tg*Dre2-mAID-HA, rHA-mAID-*Tg*Nar1, r*Tg*CIA2-mAID-HA, and r*Tg*MMS19-mAID-HA parasites cultured in the absence or presence of IAA for 24 h, probed with anti-puromycin, anti-HA or anti-*Tg*Tom40 antibodies. The membrane was also stained following transfer with the protein-binding dye Ponceau S. **(B)** Relative abundance of puromycin incorporation into each parasite line was determined as a percentage of the -IAA control, with abundances normalized using the *Tg*Tom40 loading control. Data points represent the mean±SD of three independent experiments.

Data were analyzed using a one-way ANOVA followed by Tukey's multiple comparisons test with relevant $p$ values shown. (**C**) Western blots of proteins extracted from r*Tg*Dre2-mAID-HA/*Tg*ABCE1-cMyc parasites cultured for 0, 24, or 48 h in IAA and separated by SDS-PAGE. Samples were probed with anti-HA, anti-cMyc, and anti-*Tg*Tom40 antibodies. The relative abundance of the *Tg*ABCE1-cMyc protein in the western blot was determined as a percentage of the 0 h control, with abundances normalized using the *Tg*Tom40 loading control. Data points represent the mean ± SD of three independent experiments. Data were analyzed using a one-way ANOVA followed by Tukey's multiple comparisons test, with $p$ values shown. The numerical data underlying this Figure can be found in S1 Data.
(TIF)

**S5 Fig. Multiple sequence alignment of the *Tg*CIA2 protein with homologs from other eukaryotes.** A multiple sequence alignment of the *Tg*CIA2 protein with homologs from the yeast *Saccharomyces cerevisiae* (*Sc*CIA2; UniProt accession number P38829), *Homo sapiens* (*Hs*CIA2; UniProt Q9Y3D0), and the fruit fly *Drosophila melanogaster* (*Dm*CIA2; UniProt Q9VTC4). The reactive cysteine residue of CIA2 that is proposed to function in FeS cluster binding is highlighted by a red box.
(TIF)

**S6 Fig. Characterizing the protein composition of the CIA Targeting Complex (CTC) in *Toxoplasma gondii* parasites. (A)** Western blot of proteins extracted from r*Tg*Dre2-mAID-HA, r*Tg*Tah18-mAID-HA, and rHA-mAID-*Tg*Nar1/*Tg*CIA1-smFP-cMyc parasites, separated by BN-PAGE and probed with anti-HA antibodies to detect the *Tg*Dre2-mAID-HA, *Tg*Tah18-mAID-HA, and HA-mAID-*Tg*Nar1 proteins. **(B, C)** Western blots of proteins extracted from r*Tg*CIA1-mAID-cMyc/*Tg*CIA2-HA parasites cultured for 0–12 h in IAA, separated by (**B**) SDS-PAGE or (**C**) BN-PAGE and probed with anti-HA antibodies to detect the *Tg*CIA2-HA protein, anti-cMyc antibodies to detect the *Tg*CIA1-mAID-cMyc protein, and anti-*Tg*Tom40 antibodies as a loading control. For the SDS-PAGE western blots (**B**), the relative abundance of the *Tg*CIA2-HA protein was determined as a percentage of the 0 h IAA control, with abundances normalized using the *Tg*Tom40 loading control. Data points represent the mean ± SD of three independent experiments. Data were analyzed using a one-way ANOVA followed by Tukey's multiple comparisons test with relevant $p$ values shown. The numerical data underlying this Figure can be found in S1 Data. In the BN-PAGE western blot (**C**), the black arrowhead indicates the >720 kDa candidate CIA Targeting Complex. **(D)** Western blot of proteins extracted from r*Tg*CIA2-mAID-HA/*Tg*CIA1-smFP-cMyc parasites and immunoprecipitated using anti-HA- or anti-cMyc-conjugated agarose beads. Extracted fractions include total protein prior to immunoprecipitation (T), unbound proteins (U), and antibody-bound proteins (B) in the indicated proportions. Protein fractions were separated by SDS-PAGE and probed with anti-cMyc, anti-HA, or anti-*Tg*Tom40 antibodies. Data are representative of three independent experiments for the anti-HA immunoprecipitation and two independent experiments for the anti-cMyc immunoprecipitation. Asterisks depict likely degradation products of the *Tg*CIA1-smFP-cMyc protein.
(TIF)

**S7 Fig. Epitope tagging candidate CIA pathway proteins in *Toxoplasma gondii*. (A)** A mAID-cMyc epitope tag was integrated into the 3′ region of the open reading frames of *Tg*CIA1 in RHΔ*ku80*/Tir1-FLAG/tdTomato parasites, generating the r*Tg*CIA1-mAID-cMyc parasite line. **(B, C)** HA epitope tags were integrated into the (**B**) *Tg*CIA2 or (**C**) *Tg*MMS19 loci of the r*Tg*CIA1-mAID-cMyc parasite line, generating the r*Tg*CIA1-mAID-cMyc/*Tg*CIA2-HA and r*Tg*CIA1-mAID-cMyc/*Tg*MMS19-HA parasite lines. **(D)** A spaghetti monster fluorescent protein-cMyc (smFP-cMyc) epitope tag was integrated into the 3′ region of the open reading frame of *Tg*CIA1 in IAA-regulatable r*Tg*CIA2-mAID-HA (top) or r*Tg*MMS19-mAID-HA (bottom) parasites, generating the r*Tg*CIA2-mAID-HA/*Tg*CIA1-smFP-cMyc and r*Tg*MMS19-mAID-HA/*Tg*CIA1-smFP-cMyc parasite lines. A schematic depicting the target locus before and after modification, the approximate position of the forward and reverse primers used in the PCR analysis, and the expected sizes of the PCR products in the native and modified genomic loci, are shown at the top of each panel. The PCR analyses testing for genomic modifications are shown at the

 

bottom of each panel. PCRs were performed using forward and reverse primers specific to the target site of each gene, and using genomic DNA extracted from clonal parasite lines. Genomic DNA from a WT parasite line was used as a control for the expected size of the native locus in each screen.
(TIF)

**S8 Fig. The mitochondrial localization of *Tg*CIA1 is not dependent on other candidate CIA pathway proteins. (A)** Immunofluorescence assays of r*Tg*NBP35-cMyc/*Tg*CIA1-HA parasites, cultured in the absence (top) or presence (bottom) of ATc for two days. Samples were probed with anti-HA to detect *Tg*CIA1-HA (green) and anti-*Tg*Tom40 antibodies to detect the mitochondrion (magenta). **(B)** A spaghetti monster fluorescent protein-cMyc (smFP-cMyc) epitope tag was integrated into the 3′ region of the open reading frame of *Tg*CIA1 in IAA-regulatable r*Tg*Tah18-mAID-HA (top) or rHA-mAID-*Tg*Nar1 (bottom) parasites, generating the r*Tg*Tah18-mAID-HA/*Tg*CIA1-smFP-cMyc and rHA-mAID-*Tg*Nar1/*Tg*CIA1-smFP-cMyc parasite lines. A schematic depicting the target locus before and after modification, the approximate position of the forward and reverse primers used in the PCR analysis, and the expected sizes of the PCR products in the native and modified genomic loci, are shown at the top of the panel. The PCR analyses testing for genomic modifications are shown at the bottom of the panel. Note that the PCR screens for the candidate r*Tg*Tah18-mAID-HA/*Tg*CIA1-smFP-cMyc clones was performed on the same gel as the r*Tg*CIA2-mAID-HA/*Tg*CIA1-smFP-cMyc clone (S7D Fig) and therefore has the same ladder and WT control. **(C)** Immunofluorescence assays of r*Tg*Tah18-mAID-HA/*Tg*CIA1-smFP-cMyc parasites cultured in the absence (top) or presence (bottom) of IAA for 24 h. Samples were probed with anti-cMyc to detect *Tg*CIA1-smFP-cMyc (green) and anti-*Tg*Tom40 antibodies to detect the mitochondrion (magenta). **(D)** Multiple sequence alignment of the C-terminal region of the *Toxoplasma gondii* Nar1 protein (*Tg*Nar1) with Nar1 homologs from the chrompodellid *Vitrella brassicaformis* (*Vb*Nar1; www.veupathdb.org accession number Vbra_21454; [60]), the ciliate *Tetrahymena thermophila* (*Tt*Nar1; UniProt accession number Q22NP0), the amoebozoan *Dictyostelium discoideum* (*Dd*Nar1; UniProt Q54F30), the plant *Arabidopsis thaliana* (*At*Nar1; UniProt Q94CL6), and the animals *D. melanogaster* (*Dm*Nar1; UniProt Q8SYS7) and *H. sapiens* (*Hs*Nar1; NCBI accession number NP071938.1). **(E, F)** Western blot of proteins extracted from rHA-mAID-*Tg*Nar1/*Tg*CIA1-smFP-cMyc parasites cultured in the absence or presence of IAA for 24 h. Proteins were separated by SDS-PAGE **(E)** or BN-PAGE **(F)** and were probed with anti-cMyc, anti-HA, or anti-*Tg*Tom40 antibodies. The mean relative abundance of the *Tg*CIA1-smFP-cMyc protein for the SDS-PAGE western blot was quantified a percentage of the no IAA control following normalization using the *Tg*Tom40 loading control (**E**, right). Data points represent the mean ± SD of three independent experiments and were analyzed using a paired *t* test with the *p* value shown. The numerical data underlying this Figure can be found in S1 Data. The black arrowhead indicates the >720 kDa CTC and the red and blue arrowheads indicate the lower mass *Tg*CIA1-containing complexes observed in the BN-PAGE western blot (**F**). **(G)** Western blot of proteins extracted from rHA-mAID-*Tg*Nar1/*Tg*CIA1-smFP-cMyc parasites and immunoprecipitated using anti-HA-conjugated agarose beads. Extracted fractions include total protein prior to immunoprecipitation (T), unbound proteins (U), and antibody-bound proteins (B). Protein fractions were separated by SDS-PAGE and probed with anti-cMyc, anti-HA, or anti-*Tg*Tom40 antibodies. Data are representative of three independent experiments. **(H)** Immunofluorescence assays of rHA-mAID-*Tg*Nar1/*Tg*CIA1-smFP-cMyc parasites, cultured in the absence (top) or presence (bottom) of IAA for 24 h. Samples were probed with anti-cMyc to detect *Tg*CIA1-smFP-cMyc (green) and anti-*Tg*Tom40 antibodies (magenta) to detect the mitochondrion. For all immunofluorescence assays depicted in this figure, the scale bars are 2 μm and DIC denotes the differential interference contrast transmission image.
(TIF)

**S9 Fig. Multiple sequence alignment of *Tg*CIA1 with homologs from other eukaryotes.** A multiple sequence alignment of the *Tg*CIA1 protein with homologs from the dinozoans *Perkinsus marinus* (NCBI accession number: XP_002785992.1) and *Symbiodinium microadriaticum* (NCBI CAE7872903.1), the apicomplexans *Plasmodium falciparum* (www.veupathdb.org accession number PF3D7_1209400; [60]) and *Cryptosporidium parvum* (VEuPathDB cgd1_2230),

the chrompodellids *Vitrella brassicaformis* (VEuPathDB Vbra_18342), and *Chromera velia* (VEuPathDB Cvel_5245), the yeast *Saccharomyces cerevisiae* (UniProt accession number Q05583), the ciliate *Paramecium tetraurelia* (UniProt A0BLF7), the plant *Arabidopsis thaliana* (UniProt F4JVW1), the amoebozoan *Dictyostelium discoideum* (UniProt Q55DA2), the animal *Drosophila melanogaster* (UniProt Q7K1Y4) and the oomycete *Phytophthora infestans* (UniProt D0NMX9). The positions of each of the seven blades of the β-propellor structure of CIA1 are indicated. Sequences corresponding to the CD loops of interest are highlighted in pink (CD1 loop), yellow (CD3 loop), and green (CD5 loop), with the remaining CD loops and the AB loop of blade 7 (which is slightly extended in *Tg*CIA1) highlighted in blue. The alignment was generated in Clustal Omega and plotted using pyBoxshade.
(PDF)

**S10 Fig. The effects of mutating the CD loops of *Tg*CIA1 on mitochondrial morphology and parasite proliferation.** (A) Relative abundances of proteins depicted in the Fig 6B western blots were determined as a percentage of the -IAA condition for the c*Tg*CIA1$_{WT}$-Ty1 protein and normalized using the *Tg*Tom40 loading control. Data points represent the mean ± SD of three independent experiments. Data were analyzed using a one-way ANOVA followed by Tukey's multiple comparisons test, with significant *p* values (<0.05) with respect to the -IAA condition of the c*Tg*CIA1$_{WT}$-Ty1-IAA protein shown. (B) Quantification of mitochondrial morphology in r*Tg*CIA1-mAID-HA/c*Tg*CIA1$_{WT}$-Ty1 (WT-Ty1), r*Tg*CIA1-mAID-HA/c*Tg*CIA1$_{ScCD1}$-Ty1 (*Sc*CD1-Ty1), r*Tg*CIA1-mAID-HA/c*Tg*CIA1$_{ScCD3}$-Ty1 (*Sc*CD3-Ty1), and r*Tg*CIA1-mAID-HA/c*Tg*CIA1$_{ScCD5}$-Ty1 (*Sc*CD5-Ty1) parasites. Mitochondria were observed by immunofluorescence assays using anti-*Tg*Tom40 antibodies, and were classified as lasso, branched, linear, or tadpole shaped, with representative images of each category shown above. The morphologies of mitochondria in 150 vacuoles containing 4–16 intracellular parasites were determined in each parasite line across three independent experiments, with the observer blinded to the identities of the samples being examined. (C–E) Plaque assays of r*Tg*CIA1-mAID-HA parasites and r*Tg*CIA1-mAID-HA parasites constitutively expressing *Tg*CIA1 variants, including (C) c*Tg*CIA1$_{WT}$-Ty1 (WT-Ty1), c*Tg*CIA1$_{ScCD1}$-Ty1 (*Sc*CD1-Ty1), c*Tg*CIA1$_{ScCD3}$-Ty1 (*Sc*CD3-Ty1) and c*Tg*CIA1$_{ScCD5}$-Ty1 (*Sc*CD5-Ty1), (D) c*Tg*CIA1$_{WT}$-Ty1 (WT-Ty1), c*Tg*CIA1$_{CD5-to-CD1}$-Ty1 (CD5-to-CD1-Ty1), c*Tg*CIA1$_{VbCD5}$-Ty1 (*Vb*CD5-Ty1), and c*Tg*CIA1$_{SmCD5}$-Ty1 (*Sm*CD5-Ty1), and (E) c*Tg*CIA1$_{WT}$-Ty1 (WT-Ty1), c*Tg*CIA1$_{W526A}$-Ty1 (W526A-Ty1) c*Tg*CIA1$_{Y527A}$-Ty1 (Y527A-Ty1), c*Tg*CIA1$_{F532A}$-Ty1 (F532A-Ty1), and c*Tg*CIA1$_{R533A}$-Ty1 (R533A). Parasites were cultured in the absence (top) or presence (bottom) of IAA for six days and are representative of three independent experiments. Each plaque assay was set up simultaneously with the fluorescence proliferation assays depicted in Figs 6F, 7C, 8I, 9D, and S12G. The numerical data underlying this Figure can be found in S1 Data.
(TIF)

**S11 Fig. The CD5 loop of *Tg*CIA1 is sufficient to target GFP to the mitochondrion.** Immunofluorescence assay of parasites constitutively expressing a GFP-Ty1 variant containing the CD5 loop of *Tg*CIA1 between the eighth and ninth β-strands of GFP (GFP$_{TgCD5}$-Ty1), probed with anti-Ty1 antibodies to detect the GFP$_{TgCD5}$-Ty1 protein (green) and anti-*Tg*Tom40 antibodies to detect the mitochondrion (magenta). Scale bars are 2 μm. DIC, differential interference contrast. Right, corresponding fluorescence plots depicting the intensity of anti-Ty1 (green) and anti-*Tg*Tom40 (magenta) labeling along the yellow line in merged images. The numerical data underlying this Figure can be found in S1 Data.
(TIF)

**S12 Fig. An aromatic amino acid motif in the CD5 loop of *Tg*CIA1 facilitates mitochondrial targeting.** (A) Sanger DNA sequencing chromatograms depicting the nucleotides modified in the *Tg*CIA1 gene to generate substitutions of the W526, Y527, and R533 residues of the protein to alanine. Mutated codons for each line are highlighted with a magenta box. (B) The relative abundance of proteins depicted in Fig 8B was determined as a percentage of the WT-HA protein and normalized using the *Tg*Tom40 loading control. Data points represent the mean ± SD of three independent

experiments. Data were analyzed using a one-way ANOVA followed by Tukey's multiple comparisons test with $p$ values shown. **(C, D)** Western blots of proteins extracted from r*Tg*CIA1-mAID-HA/c*Tg*CIA1_{WT}-Ty1 (WT-Ty1), r*Tg*CIA1-mAID-HA/c*Tg*CIA1_{W526A}-Ty1 (W526A-Ty1), r*Tg*CIA1-mAID-HA/c*Tg*CIA1_{Y527A}-Ty1 (Y527A-Ty1), r*Tg*CIA1-mAID-HA/c*Tg*CIA1_{F532A}-Ty1 (F532A-Ty1), and r*Tg*CIA1-mAID-HA/c*Tg*CIA1_{R533A}-Ty1 (R533A-Ty1) parasites, separated by **(C)** SDS-PAGE or **(D)** BN-PAGE, and probed with anti-Ty1 or anti-*Tg*Tom40 antibodies. The black arrowhead indicates the >720 kDa CIA Targeting Complex; red and blue arrowheads indicate the lower mass complexes containing the *Tg*CIA1 protein. **(E)** Immunofluorescence assays of r*Tg*CIA1-mAID-HA/c*Tg*CIA1_{WT}-Ty1 (WT-Ty1; also shown in Fig 8G), r*Tg*CIA1-mAID-HA/c*Tg*CIA1_{W526A}-Ty1 (W526A-Ty1), r*Tg*CIA1-mAID-HA/c*Tg*CIA1_{Y527A}-Ty1 (Y527A-Ty1), and r*Tg*CIA1-mAID-HA/c*Tg*CIA1_{R533A}-Ty1 (R533A-Ty1) parasites. The complemented proteins of interest (green) and the mitochondrion (magenta) were labeled with anti-Ty1 and anti-*Tg*Tom40 antibodies, respectively. Schematics depicting the modified amino acid sequence in the CD5 motif of the proteins from each panel are included next to images (left). Scale bars are 2 µm. DIC, differential interference contrast. Right, corresponding fluorescence profile depicting intensity of anti-Ty1 (green) and anti-*Tg*Tom40 (magenta) labeling along the yellow lines of the merged images. **(F)** The correlation between Ty1-tagged proteins and *Tg*Tom40 was quantified using the Pearson correlation coefficient ($r$) and the data were analyzed using a one-way ANOVA followed by Tukey's multiple comparisons test with $p$ values shown. **(G)** Fluorescence proliferation assays of r*Tg*CIA1-mAID-HA and r*Tg*CIA1-mAID-HA/c*Tg*CIA1_{WT}-Ty1 (also shown in Fig 8I), r*Tg*CIA1-mAID-HA/c*Tg*CIA1_{W526A}-Ty1, r*Tg*CIA1-mAID-HA/c*Tg*CIA1_{Y527A}-Ty1, and r*Tg*CIA1-mAID-HA/c*Tg*CIA1_{R533A}-Ty1 parasites, grown in the absence (black) or presence (red) of IAA. Parasite proliferation is expressed as a percentage of the fluorescence measurement in the -IAA condition on the final day of the assay for each line. Individual data points and error bars represent the mean ± SD of three technical replicates. Error bars not visible are smaller than the symbol. Data are representative of three independent experiments. The numerical data underlying this Figure can be found in S1 Data.
(TIF)

**S13 Fig. Immunofluorescence assay controls on untagged parasites. (A–C)** Immunofluorescence assays of TATiΔ*ku80* parasites lacking epitope tags (**A–C**, top) and epitope-tagged positive control parasites (**A, B** bottom, *Tg*CIA2-mAID-HA/*Tg*CIA1-smFP-cMyc parasites; **C** bottom, *Tg*CIA1-mAID-HA/c*Tg*CIA1_{WT}-Ty1 parasites) were performed to test for non-specific antibody labeling in untagged parasites. Parasite samples were probed on the same day with **(A)** anti-HA (green) and anti-*Tg*Tom40 (magenta) antibodies, **(B)** anti-cMyc (green), and anti-*Tg*Tom40 (magenta) antibodies, and **(C)** anti-Ty1 (green) and anti-*Tg*Tom40 (magenta) antibodies using the same antibody dilutions for the negative and positive control samples. For image processing, the contrast and brightness of a positive control image of each sample was adjusted linearly, and the same adjustments (*i.e.,* the same minimum and maximum pixel intensities) were applied to TATiΔ*ku80* parasites. Scale bars are 2 µm. DIC, differential interference contrast transmission image.
(TIF)

**S1 Table. Oligonucleotides and gBlocks used in this study (all oligonucleotides are listed in a 5′ to 3′ orientation; sgRNA-coding sequences are underlined).**
(PDF)

**S1 Raw images. The source images of all the western blots, PCR gels, and plaque assays that were cropped in generating figures in the manuscript.** Refer to the Figure legends in the manuscript for details on each figure.
(PDF)

**S1 Data. The raw and normalized numerical data for the graphical figures depicted in the manuscript.** Refer to the Figure legends in the manuscript for details on each figure.
(XLSX)

**S2 Data. Replicate data of representative experiments included in the manuscript, including immunofluorescence assays, SDS-PAGE and BN-PAGE western blots, fluorescence proliferation assays, and plaque assays.** Refer to the Figure legends in the manuscript for details on each figure.
(PDF)

## Acknowledgments

We thank Mick Devoy, Fei-Ju Li, and Harpreet Vohra for assistance with flow cytometry, and the 2020 ANU Cell Biology course for undertaking the initial studies on the subcellular localizations of the various CIA pathway proteins. We are grateful to VEuPathDB for providing numerous data sets and bioinformatic search tools that were integral to the research undertaken in this study.

## Author contributions

**Conceptualization:** Evie R. Hodgson, Jenni A. Hayward, Giel G. van Dooren.

**Data curation:** Evie R. Hodgson.

**Formal analysis:** Evie R. Hodgson, Jenni A. Hayward, Giel G. van Dooren.

**Funding acquisition:** Jenni A. Hayward, Rachel A. Leonard, Giel G. van Dooren.

**Investigation:** Evie R. Hodgson, Jenni A. Hayward, Rachel A. Leonard, Fadzai Victor Makota, Giel G. van Dooren.

**Methodology:** Evie R. Hodgson, Jenni A. Hayward, Fadzai Victor Makota, Giel G. van Dooren.

**Project administration:** Evie R. Hodgson, Giel G. van Dooren.

**Resources:** Giel G. van Dooren.

**Supervision:** Jenni A. Hayward, Giel G. van Dooren.

**Validation:** Evie R. Hodgson, Jenni A. Hayward, Rachel A. Leonard, Giel G. van Dooren.

**Visualization:** Evie R. Hodgson, Jenni A. Hayward, Rachel A. Leonard, Giel G. van Dooren.

**Writing – original draft:** Evie R. Hodgson, Giel G. van Dooren.

**Writing – review & editing:** Evie R. Hodgson, Jenni A. Hayward, Fadzai Victor Makota, Giel G. van Dooren.

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
