## [Editor Report · Decision Letter 0]

5 Feb 2025

Dear Giel,

Thank you for submitting your manuscript entitled "The mitochondrion is a hub for the cytosolic iron-sulfur cluster assembly pathway in Toxoplasma parasites" for consideration as a Research Article by PLOS Biology. Happy to see it here.

As I mentioned by e-mail during the presubmission, I am writing to let you know that we would like to send your submission out for external peer review.

Once your full submission is complete, your paper will undergo a series of checks in preparation for peer review. After your manuscript has passed the checks it will be sent out for review. To provide the metadata for your submission, please Login to Editorial Manager (https://www.editorialmanager.com/pbiology) within two working days, i.e. by Feb 07 2025 11:59PM.

Best wishes,

Melissa

Melissa Vazquez Hernandez, Ph.D.

Associate Editor

PLOS Biology

---

## [Decision Letter · Decision Letter 1]

17 Mar 2025

Dear Giel,

I hope all is well. Thank you for your patience while your manuscript "The mitochondrion is a hub for the cytosolic iron-sulfur cluster assembly pathway in Toxoplasma parasites" was peer-reviewed at PLOS Biology. It has now been evaluated by the PLOS Biology editors, an Academic Editor with relevant expertise, and by three independent reviewers.

In light of the reviewers' comments, which are included at the end of this email, we would like to invite you to revise your manuscript to fully address the concerns raised in the reports. As you will see, the majority of reviewers expressed a positive view regarding the relevance and novelty of the study. However, several concerns—primarily related to quantification—were highlighted and will need to be addressed through additional experimental work. Reviewer 1 raised concerns about the support for certain conclusions and recommended additional methods, such as immunoblotting or calculating Pearson correlation coefficients, to conclusively determine the localization of specific components. They also requested that you assess the impact of translation on some of the mutants. Reviewer 2 provided minor suggestions pertaining to figure presentation. Reviewer 3 mentioned that some controls might be missing and noted that the conclusion regarding mitochondrial localization being essential for survival is not fully supported by the current data. Like Reviewer 1, this reviewer also pointed out potential issues with your quantification methods, which may contribute to inconsistencies in the results. We concur with the reviewers’ concerns and request that you revise the manuscript accordingly, particularly addressing the quantification issues, which we believe are critical for strengthening the study.

Given the extent of revision needed, we cannot make a decision about publication until we have seen the revised manuscript and your response to the reviewers' comments. Your revised manuscript is likely to be sent for further evaluation by all or a subset of the reviewers.

**IMPORTANT - SUBMITTING YOUR REVISION**

*Re-submission Checklist*

*Published Peer Review*

*PLOS Data Policy*

*Blot and Gel Data Policy*

Sincerely,

Melissa

Melissa Vazquez Hernandez, Ph.D.

Associate Editor

PLOS Biology

REVIEWERS' COMMENTS

Reviewer #1 (Sébastien Besteiro):

In this manuscript Hodgson et al present compelling evidence that the mitochondrion is used as a platform for the cytosolic iron-sulfur assembly pathway in the parasite Toxoplasma gondii. The demonstration of this mitochondrion-located assembly pathway is very original, and at the same time it makes complete sense functionally, as the CIA pathway is likely provided with a sulfur-precursor for Fe-S cluster assembly that originates from the mitochondrion.

There is a tremendous amount of work presented in this manuscript and the authors have analyzed extensively the sub-cellular localization and the respective contribution of the different components of the CIA pathway to the overall fitness of the parasite.

In my opinion, the authors have unambiguously demonstrated:

- that each component of the pathway (TAH18, NAR1, CIA1, CIA2, MMS19) is important for translation and essential for parasite fitness.

- that components of the pathway either localize to the mitochondrion (TAH18, NAR1), or have a dual cytosolic/mitochondrial localization (MMS19, CIA1, CIA2).

- that the targeting of the CIA targeting complex (CIA) is primarily mediated by protein CIA1, with a particular contribution of a myzozoan-specific amino acid loop.

So, I have no doubt about the novelty and interest of the research described in the manuscript. The experiments are sound and in general well-controlled. However, in my opinion there are a few conclusions that are not completely substantiated by the data and would deserve a more clear-cut investigation.

First, as a readout for mitochondrial, cytosolic or dual localization, the authors mainly use representative images and pixel intensity profiles. Yet, there are a number of experiments for which, as stated by the authors themselves several times in the manuscript, they « cannot rule out that some mitochondrial localization still occurs » and these images are indeed not very conclusive. Perhaps as a result, there are some discrepancies between their conclusion on the localization of some constructs and the impact (or lack of impact) on the parasites. Typical examples include the single amino acid mutants on Fig. 8D and Fig. 9A&B, in which the CIA1 signal still show some extensive overlap with the mitochondrion. I think that a less subjective (and easier to quantify) way of assessing the distribution of the proteins would be through an analysis by immunoblot of protein extracts after mechanical disruption of the parasites and quantifying the relative organelle-associated and cytosolic fractions (ie membrane-associated and soluble forms of the proteins). Alternatively, providing Pearson correlation coefficient values when comparing the signals for the proteins of interest and the mitochondrion on a large number of parasites might allow a more direct comparison and provide more compelling evidence. Again, this is particularly important for data interpretation for a number of fitness phenotypes. For the single amino acid mutants for example, Fig. 8 shows that some are not impacted for fitness, while their localization is currently presented in the manuscript as being largely altered. Similarly, it may be difficult to conclude on the localization of the SmCD5CIA1 and VbCD5CIA1 cell lines are viable in spite of a potential mislocalization (Fig. 9C), although images suggest that both proteins might still be present at the mictochondrion (Fig. 9A&B). Finally, the CIA1ScCD5 cell line is presumably « cytosolic » and severely impacted in growth (Fig. 6), yet translation seems to proceed normally in these parasites (Fig. 10B), which is very surprising. Overall, more clear-cut and quantified data on protein localization may help resolve some of these discrepancies.

Second, while there seems to be a marked impact of CIA1 depletion on protein translation (Fig. 3C), it seems to be less obvious when depleting the other members of the complex (S4 Fig.). First, there it seems unlikely that these analyses have been performed 24h after adding the IAA (as stated in the figure legends, l. 1174 and l. 1400), because at this timepoint the authors have clearly shown that the decrease in abundance of the ABCE1 protein (which is presumably responsible for the phenotype) only happens at 48 h post IAA addition (Fig. 3A). One could imagine that not all CIA assembly proteins are impacting ABCE1 with exactly the same kinetics and it would have been informative to look for ABCE1 depletion in the mutant cell lines. While I realize it is quite tedious to do so for all the mutant cell lines (Tah18, Der2, Nar1, Cia2, MMS19), I think it would be a priority for the Dre2 mutant for which, quite unexpectedly, no impact on translation is observed. Assessing the impact of Der2 depletion on ABCE1 abundance, perhaps including longer IAA incubation times, would provide a more direct and clearer evidence that Dre2 depletion is not affecting the stability of ABCE1.

Minor points.

Please provide scale bars for all series of microscopy pictures.

The authors seem to have chosen the yeast nomenclature for naming the different components of the complex (although MMS19 is also called MET18 in yeast). As these proteins have different names in human and plant models (a nomenclature chosen by other Toxoplasma researchers on the basis that many Fe-S proteins are common with plants due to the presence of a plastid), perhaps mention the protein aliases somewhere in the manuscript so that readers could relate more easily with the existing literature?

l. 137-140: the accessibility of the proteins to the degron system is another hint that they are exposed to the cytosolic face of the mitochondrion. That may be briefly mentioned.

Fig. 4B. The impact of CIA1 depletion on the abundance of other CTC proteins is variable. Perhaps the authors could very briefly discuss that it may be explained by a different stability of these proteins.

Fig. 4A/4F. Any hint as to why the intensity of labeling of CIA1 in the low molecular weight complexes vs the high molecular band seem inconsistent between these experiments (same for Fig. 6D)?

l. 308-309. The authors say that the absence of co-immunoprecipitation between CIA1 and NAR1 suggests that they are likely not interacting. The native gel data and the way CIA1 depletion affects CIA2 and MMS19 localization are solid and converging evidence that they would interact, but as cell lines with tagged proteins of interest have been generated, have the authors performed co-immunoprecipitation experiments between the CTC proteins that would further support their interaction within the complex?

More emphasis may be put in the discussion on the interesting finding that segregating CIA1 to the mitochondrion severely impacts translation, which implies the need for some shuttling and perhaps more local interaction with cytosolic components for Fe-S cluster transfer. On the other hand, as mentioned above, I am not completely convinced by the lack of impact of a "cytosol only"-localized version of the protein, for the simple reason that the microscopic imaging data currently presented does not allow to completely rule out that some protein is at the mitochondrion.

Reviewer #2:

* Summary:

This group aims to define the CIA (cytosolic iron sulfur cluster biosynthesis pathway) in apicomplexan parasite Toxoplasma gondii. The group took interest in exploring this pathway further when they previously identified a component, NBP35 (CIA scaffold protein) was localized unexpectedly to the mitochondria outer membrane instead of the cytosol. This led them to determine if other components of the pathway were also found at the surface of the mitochondrion or if they remained in the cytosol like many conserved CIA proteins in other eukaryotes. Surprisingly, many of their candidate CIA proteins involved in forming the CIA Targeting Complex (CTC) showed dual localization to the mitochondria outer membrane and the cytosol and were found essential for parasite growth. Further, they identified a unique structural component of one of the CTC proteins, belonging to CIA1, that allows it to localize to the mitochondria membrane by an aromatic amino acid motif within a loop of the protein. This was identified using the AlphaFold2 structural prediction for the protein and comparing it to CIA1 from other eukaryotes. This novel loop structure is specifically conserved in myzozoans, which would lead them to believe other organisms in this clade, as well as other apicomplexan parasites may also show a dual localization of these proteins to the mitochondria and cytosol. This interaction between the CTC and T. gondii mitochondria was found to be essential in facilitating Fe-S cluster synthesis and viability of the parasite.

Strengths of this manuscript:

1. Identification of key residues for proper protein function in the pathway, then elegantly showing that mutating said residues reverses the phenotype.

2. Identification of structural components of the proteins that bind to the mitochondria outer membrane- then determining if this is a conserved feature. Extensive experiments that exhaustively show the importance of the CD5 loop structure of CIA1 and its role in the location of the protein at the mitochondria outer membrane.

3. Only minor modifications needed for the presentation of the data in figures.

4. Demonstrated that other myzozoan CD5 loop sequences complemented the phenotype in the knockdown line for CIA1 . This demonstrated the function of this loop is actually conserved and rescues growth and localization.

* Major points to address:

Figure 3B, 3D, 4C, 10C: The data are presented as % control making it hard to appreciate the extent of variation In the control. There are two options to plot variation in the control group. First, generate an average value for the control and then plot each control replicate as a % of this average. Alternatively, show the non-normalized data on the supplement.

Figure 8B: If making the claim that the protein abundance of the amino acid mutants are similar to WT, please include a graph with the protein abundance values below.

The authors speculate that association of the CIA pathway arose out of need to efficiently capture iron, during evolution of the myzozoan ancestor. However, it is not clear why localizing the CIA proteins to the outer member would offer much advantage over simple diffusion given the small cytosolic compartment of the parasite cell. Doesn't this imply that direct interaction between CIA proteins in the outer surface of the mitochondrion with FeS cluster clients is required and may be the reason for the importance of the association? Figure 10 alludes to this, but other more speculative alternatives are also offered. It would help to provide more clarity on what the authors think is the mostly likely explanation and why.

* Minor points to address:

Line 21: 'clusters' not 'cluster'

Line 103, 108, etc: Authors do not need to refer readers to PCR verification of strains utilized in every figure. Citation is enough referencing supplemental figures is enough.

Figure 1F: Western blot of tagged MMS19-mAID-HA shows two major bands. Are both bands corresponding to the protein? (one cleaved and one un-cleaved). If not, which is considered the protein of interest?

Figure 6E: Move the color-coded key for -IAA / + IAA to the bottom so it's clear it goes with all graphs in E.

Figure 6E: There appears to be a slight fitness defect for parasites with the ScCD1 mutation- is there any speculation as to why this confers a fitness defect but not a localization difference of CIA1?

-All figures containing growth curves: Can statistical values be plotted on the graph?

Figure 8E: move color coded key to a more visible location. Statistical values plotted would also be great.

Reviewer #3:

In this study, the authors conduct a detailed analysis of the cytosolic iron-sulfur cluster assembly (CIA) pathway in Toxoplasma gondii. The authors demonstrate the essentiality of this pathway for parasites, as conditional knockdown of CIA genes prevents parasite growth. Interestingly, while CIA proteins are typically found in the cytosol of other previously examined eukaryotes, the authors find that many of the CIA components in T. gondii have exclusive or dual localizations to the mitochondrion. The authors then focus on the CIA targeting complex (CTC), generating many mutant and chimeric versions of CTC proteins to examine the function of specific domains. The authors find that CIA1 is required for trafficking of other CTC components to the mitochondrion, and that the CD5 loop of CIA1 in particular is required for this localization.

The authors have generated an impressive number of strains to dissect the localization and function of the CTC, and examine the divergent parasite CTC within a broader evolutionary context. While much of the data is well-presented, many experiments are missing essential controls. Additionally, a major claim of the paper - that the mitochondrial localization of the CTC is critical for parasite survival - is directly disputed by the authors' own data. These and other minor concerns should be address prior to publication.

Major Comments:

1. A major claim of the paper is that the mitochondrial localization of the CTC is required for parasite survival. However, the authors' data indicates otherwise. Fig. 8D-E show CIA1 mutants that have cytosolic localization, yet still result in viable parasites. Additionally, Fig. 10B-C demonstrates that a cytosolically-localized mutant of CIA1 is able to sustain translation, whereas a mitochondrially-localized mutant of CIA1 is not. The authors present this translation result as surprising, but it agrees with their earlier finding in Fig. 8 that cytosolic CIA mutants remain viable. Together, these results contradict the authors' statement that "the mitochondrial localization of the CTC is critical for parasite survival".

2. The manuscript relies heavily on immunofluorescence assays. However, in no instance do the authors provide an image of an untagged control strain in their experiments. Background signal is a common occurrence in immunofluorescence imaging, and could be contributing signal to colocalization quantifications. For this reason, untagged controls must be included for all immunofluorescence assays.

3. Additionally, the authors show and quantify a single vacuole for many of their immunofluorescence assays. It would benefit the manuscript if the authors could comment on the penetrance of the phenotypes they observed. This would be particularly needed in instances in which the authors claim a subtle mitochondrial colocalization despite a dominant cytosolic signal, such as in Fig. 9B.

4. The authors calculate colocalization signal along a straight line through the parasites. However, the parasite nucleus is relatively large, and will exclude signal from both the cytosol and the mitochondrion. For this reason, straight lines that cross through the nucleus are likely to artificially inflate colocalization measures. The authors should consider other measures for calculating colocalization, such as simply determining the total overlapping area of two signals, or at a minimum comment on the impact of the nucleus on their colocalization metric.

5. The authors claim that CIA1 depletion both results in the depletion of CIA2 and a lack of mitochondrial localization of CIA2. These results seem inconsistent with each other. Indeed, there appears to be no obvious decrease in CIA2-HA signal in Fig. 5A upon CIA1 knockdown, inconsistent with the western blot results in Fig. S6B showing CIA2-HA depletion. This discrepancy highlights the issues that arise from the above concerns that the authors show a single vacuole for most immunofluorescence assays, and fail to include untagged controls that would enable one to observe background signal.

Minor Comments:

1. The authors do not describe how they identified the parasite candidate homologs of the CIA proteins tagged in Fig. 1. Given the reported divergent localization of many of these CIA proteins, it would be beneficial to provide some detail on how these homologs were assigned.

2. It may simply be the resolution of the provided figures, but in many cases the yellow line used for colocalization measurements is difficult to see on the immunofluorescence images.

3. It is customary to place figure panels in the order in which they appear in the text. As Fig. 2B-G are discussed in the text before Fig. 2A, the authors should consider moving panel 2A to the end of the figure.

---

## [Decision Letter · Decision Letter 2]

24 Oct 2025

Dear Giel,

Thank you for your patience while we considered your revised manuscript "The mitochondrion is a hub for the cytosolic iron-sulfur cluster assembly pathway in Toxoplasma parasites" for publication as a Research Article at PLOS Biology. This revised version of your manuscript has been evaluated by the PLOS Biology editors, the Academic Editor and two of the original reviewers.

Based on the reviews and on our Academic Editor's assessment of your revision", we are likely to accept this manuscript for publication, provided you satisfactorily address the remaining points raised by Reviewer 1. In addition, the Academic Editor has requested that the immunofluorescence negative control data in R2R (reviewer 3, major comment 2) be provided as a new supplemental figure. Please also make sure to address the following data and other policy-related requests.

1) We routinely suggest changes to titles to ensure maximum accessibility for a broad, non-specialist readership, and to ensure they reflect the contents of the paper. In this case, we would suggest a minor edit to the title, as follows. Please ensure you change both the manuscript file and the online submission system, as they need to match for final acceptance:

"Mitochondrial targeting of the cytosolic iron–sulfur assembly machinery is necessary for parasite survival in Toxoplasma gondii"

2) Please add the link of the funding agencies in the Financial Disclosure statement in the manuscript details.

3) Please note that per journal policy, the model system and species studied should be clearly stated in the abstract of your manuscript (Toxoplasma gondii).

4) Thank you for providing all the raw values for the figures. We noticed that perhaps the values of 9C or 9D might be mislabeled in the tab for Fig 7.

5) Please cite the location of the data clearly in all relevant main and supplementary Figure legends, e.g. “The data underlying this Figure can be found in S1 Data” or “The data underlying this Figure can be found in https://doi.org/10.5281/zenodo.XXXXX”

6) Supplementary files (e.g., excel). Please ensure that all data files are uploaded as 'Supporting Information' and are invariably referred to (in the manuscript, figure legends, and the Description field when uploading your files) using the following format verbatim: S1 Data, S2 Data, etc. Multiple panels of a single or even several figures can be included as multiple sheets in one excel file that is saved using exactly the following convention: S1_Data.xlsx (using an underscore).

7) Please make sure that all figures use a colorblind-friendly palette.

8) Thank you for providing the original, uncropped and minimally adjusted images supporting all blot and gel results. Please rename the pdf file to ‘S1_raw_images’.

9) Please ensure that your Data Statement in the submission system accurately describes where your data can be found and is in final format, as it will be published as written there

We expect to receive your revised manuscript within two weeks.

*Published Peer Review History*

*Press*

Sincerely,

Melissa

Melissa Vazquez Hernandez, Ph.D.

Associate Editor

PLOS Biology

REVIEWERS' COMMENTS:

Reviewer #1:

I applaud the authors for providing additional data and analyses, which have considerably strengthened the manuscript. I just have two remaining comments on minor points.

- l. 68. and Fig. 9D. Are Perkinsozoa considered 'dinoflagellates' per se, or are they more precisely a sister lineage to Dinoflagellata, occupying an early branching phylogenetic position within the Dinozoa? This distinction doesn't alter the fact that they are myzozoans and, therefore, has no impact on the message of the paper, but it appears to be presented sometimes in a confusing way in the literature.

- l. 600-603. The impact of TgDre2 depletion on TgABCE1 abundance, but not on overall protein synthesis, presents contradictory phenotypes. This suggests that, although TgABCE1 is expressed at lower levels, it remains functional and likely retains its Fe-S clusters. However, as noted in the discussion section, further assays (such as 55Fe incorporation and specific immunoprecipitation/quantification) would be needed to confirm this. The question then arises: why is TgABCE1 abundance affected by TgDre2 depletion if the latter is not involved in the assembly of cytosolic Fe-S clusters? In the case of TgCIA1 depletion, the strong impact on translation (evaluated after 24h of IAA treatment, Fig. 3C) appears before there is a strong decrease in TgABCE1 abundance (mostly seen at 48h of IAA treatment, Fig. 3A) : this suggests that the impact on TgABCE1 Fe-S cluster maturation is rapid and precedes the potential lack of stability and the degradation of the apo form that subsequently would lead to a decrease in protein abundance. The effect of TgDre2 depletion on TgABCE1 abundance appears to be delayed and less pronounced compared to those observed TgCIA1 depletion, but then it is intriguing that there would be a decrease in TgABCE1 abundance at 24h of IAA treatment while there is no impact on translation (Fig. S4C). This doesn't really fit with TgABCE1's Fe-S cluster assembly being affected prior to an impact on protein stability (if relying on the TgCIA1 data). This is just a comment here, there is no need to necessarily modify the discussion but, if anything, it shows that it is still a very open question.

Reviewer #2:

The authors have adequately addressed my concerns and I have no further issues with the content of the manuscript. It will be a valuable addition to the field.

---

## [Editor Report · Decision Letter 3]

10 Nov 2025

Dear Giel,

Apologies for the delay. Thank you for the submission of your revised Research Article "A novel targeting domain directs essential components of the cytosolic iron-sulfur cluster assembly pathway to the mitochondrion of Toxoplasma parasites" for publication in PLOS Biology. On behalf of my colleagues and the Academic Editor, Gary E. Ward, I am pleased to say that we can in principle accept your manuscript for publication, provided you address any remaining formatting and reporting issues. These will be detailed in an email you should receive within 2-3 business days from our colleagues in the journal operations team; no action is required from you until then. Please note that we will not be able to formally accept your manuscript and schedule it for publication until you have completed any requested changes.

IMPORTANT: we would really like to encourage you to opt for Open Peer Review, as we think that the information provided in the response to reviewers, specially regarding R1's comments, could be also informative for the readers.

PRESS

Sincerely, 

Melissa

Melissa Vazquez Hernandez, Ph.D., Ph.D.

Associate Editor

PLOS Biology
